# Switchable organoplatinum metallacycles with high quantum yields and tunable fluorescence wavelengths

Jun-Long Zhu[1], Lin Xu[1], Yuan-Yuan Ren[1], Ying Zhang[2], Xi Liu[1], Guang-Qiang Yin[1,3], Bin Sun[1], Xiaodan Cao [4], Zhuang Chen[4], Xiao-Li Zhao[1], Hongwei Tan[2], Jinquan Chen [4], Xiaopeng Li [3] & Hai-Bo Yang[1]

The preparation of fluorescent discrete supramolecular coordination complexes (SCCs) has attracted considerable attention within the fields of supramolecular chemistry, materials science, and biological sciences. However, many challenges remain. For instance, fluorescence quenching often occurs due to the heavy-atom effect arising from the Pt(II)-based building block in Pt-based SCCs. Moreover, relatively few methods exist for tuning of the emission wavelength of discrete SCCs. Thus, it is still challenging to construct discrete SCCs with high fluorescence quantum yields and tunable fluorescence wavelengths. Here we report nine organoplatinum fluorescent metallacycles that exhibit high fluorescence quantum yields and tunable fluorescence wavelengths through simple regulation of their photoinduced electron transfer (PET) and intramolecular charge transfer (ICT) properties. Moreover, 3D fluorescent films and fluorescent inks for inkjet printing were fabricated using these metallacycles. This work provides a strategy to solve the fluorescence quenching problem arising from the heavy-atom effect of Pt(II), and offers an alternative approach to tune the emission wavelengths of discrete SCCs in the same solvent.

[1] Shanghai Key Laboratory of Green Chemistry and Chemical Processes, School of Chemistry and Molecular Engineering, East China Normal University, 3663N. Zhongshan Road, Shanghai 200062, China. [2] College of Chemistry, Beijing Normal University, Beijing 100050, China. [3] Department of Chemistry, University of South Florida, Tampa, FL 33620, USA. [4] State Key Laboratory of Precision Spectroscopy, East China Normal University, Shanghai 200062, China. Correspondence and requests for materials should be addressed to L.X. (email: lxu@chem.ecnu.edu.cn) or to H.-B.Y. (email: hbyang@chem.ecnu.edu.cn)

O ver the past three decades, coordination-driven self-assembly, which is based on metal–ligand coordination interaction, has evolved to be a well-established methodology for constructing supramolecular coordination complexes (SCCs) such as one-dimensional (1-D) helices, two-dimensional (2-D) polygons, and even three-dimensional (3-D) polyhedra[1–10]. A variety of elegant and sophisticated SCCs have been fabricated through coordination-driven self-assembly, which have displayed extensive applications in host–guest chemistry[11–13], sensing[14–16], catalysis[17–21], smart polymeric materials[22,23], and biomedicines[24–26]. Recently, the preparation of fluorescent discrete SCCs has garnered great attention owing to their promising applications such as chemical sensors, optical devices, supramolecular biomedicines, and so on[27–37]. Moreover, the presence of chromophores in SCCs allows for real-time monitoring the self-assembly process and dynamics of the resultant SCCs by highly sensitive fluorescence technique[38]. Generally, two main methodologies have been explored for constructing fluorescent SCCs. One methodology is the direct introduction of chromophore as the core skeleton of the discrete SCCs[39,40]. For instance, Mukherjee et al. reported the construction of a hexagonal fluorescent SCC through the self-assembly of porphyrin-based tetratopic donor (5,10,15,20-tetrakis(4-pyridyl)porphyrin) and 90° Pt (II)-based acceptor, which displayed high encapsulating efficiency toward $Zn^{2+}$[39]. The other methodology is the preparation of discrete fluorescent SCCs by encapsulating fluorescent dyes inside the cavity of SCCs[41,42]. For example, Yoshizawa and coworkers constructed a series of fluorescent SCCs upon encapsulation of various fluorescent dyes such as BODIPY or coumarin derivatives by using a supramolecular coordination capsule[41].

Although fruitful achievements have been gained in the development of fluorescent discrete SCCs, many problems are still far from being fully resolved in this area. For example, the heavy-atom effect arising from the Pt(II)-based building block sometimes induces a significant increase in the amounts of intersystem crossing (ISC) owing to spin-orbit coupling, which thus leads to an obvious fluorescence quenching of the discrete SCCs[43,44]. Therefore, compared with their precursor building blocks, the fluorescence quantum yields of the discrete organoplatinum SCCs are relatively lower, which is usually unfavorable for their practical application. Second, there is a relative lack of efficient and simple strategy to tune the emission wavelength of the discrete SCCs[27,45]. It should be noted that the fluorescent materials with tunable emission wavelength have versatile applications as photovoltaics, light-emitting diodes, nonlinear optical materials, chemical sensors, and biological labels[46–48]. Up to now, changing the solvents is the most used method to tune the emission wavelength of the discrete SCCs since the emission wavelength of many fluorophores is often sensitive to the

solvents. However, some solvents such as acetonitrile could disrupt the Pt−N coordination bond because these solvents feature the stronger binding ability to platinum atom than pyridine[38]. Thus, in some cases, such strategy of changing the solvents exhibits certain limitation in tuning the emission wavelength of the discrete SCCs. Therefore, it is still challenging to construct discrete organoplatinum SCCs with high fluorescence quantum yields and tunable fluorescence emission wavelengths.

As two kinds of classic recognition and sensing mechanisms, photoinduced electron transfer (PET) and intramolecular charge transfer (ICT) have been widely exploited for detection of cations, anions, and small-molecules[49,50]. A typical PET molecule often includes three parts: a fluorophore that acts as the electron acceptor, a receptor that serves as an electron donor or a quencher, and a spacer that links the two parts of fluorophore and receptor (Supplementary Fig. 1–2). Fluorescent molecules based on PET are often structured as fluorophore-spacer-receptor constructs. In the PET system, the photoinduced electron transfer from the receptor to the fluorophore will induce fluorescence quenching. However, this PET process is restricted when the receptor binds upon its electron-withdrawing targets (such as metal ions), which thus induces the enhancement of fluorescence emission. However, fluorescent molecules on the basis of ICT are featured by conjugation of an electron-donating unit to an electron-accepting unit in one molecule to rise a "push–pull" π-electron system in the excited state (Supplementary Fig. 3). When the electron-accepting part interacts with an electron-withdrawing guest (such as metal ions), the electron-accepting character of the fluorescent molecule increases, thus generating a red shift in the emission spectrum. In contrast, an evident blue shift can be observed when the ICT becomes less developed owing to the interaction of the electron-donating part with an electron-withdrawing guest. Although PET and ICT mechanisms have been widely used to develop diverse sensors, probes, and molecular machine[51–54], systematic investigation and development of fluorescent SCCs by using PET and ICT has not been reported yet. By taking the inherent advantages of PET and ICT, we envision that the construction of self-assembled organoplatinum metallacycles with high fluorescence quantum yields and tunable fluorescence wavelengths could be realized by means of reasonable PET and ICT strategies.

Herein, nine triarylamine-based dipyridyl building blocks with electron-withdrawing or electron-donating groups para to the tertiary amine core (**L1**–**L9**) were reasonably designed and synthesized (Fig. 1). Interestingly, the PET and ICT properties of these nine triarylamine-based dipyridyl building blocks could be switched based on the push–pull electronic effect of the substituents. Therefore, through coordination-driven self-assembly of these nine triarylamine-based dipyridyl building blocks with

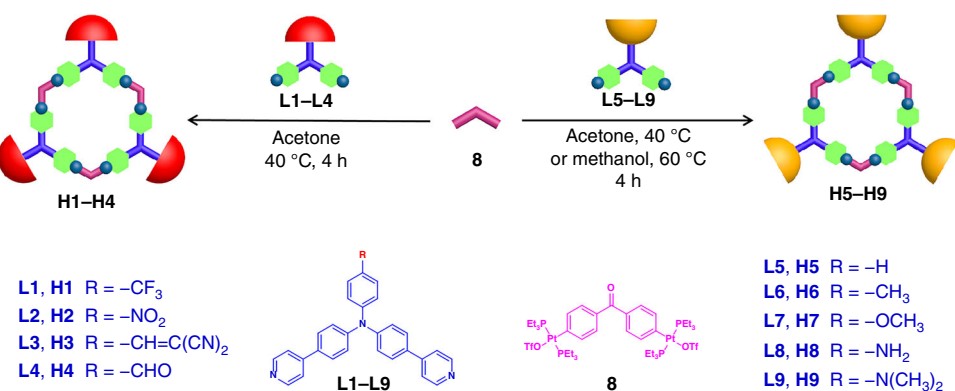

**Fig. 1** Self-assembly of 120° triarylamine-based dipyridyl donor ligands **L1**-**L9** and 120° di-Pt(II) acceptor **8** into hexagonal metallacycles **H1**-**H9**

di-Pt(II) acceptor, nine fluorescent metallacycles with tunable fluorescence wavelengths (from 480 nm to 590 nm) in the same solvent and high fluorescence quantum yields (up to 61%) were easily prepared. In this study, we presented the example on the preparation of organoplatinum metallacycles with high quantum yields and tunable fluorescence wavelengths by simply switching the PET and ICT properties of building blocks based on the substituent effect. This PET/ICT switchable strategy not only offers a good guidance on how to avoid the heavy-atom effect arising from the platinum ion in the construction of fluorescent platinum(II)-pyridyl SCCs, but also provides an alternative approach to prepare fluorescent platinum(II)-pyridyl SCCs with high quantum yields and even tunable wavelength of the discrete SCCs without changing the solvents.

## Results

**Synthesis and characterization.** With the aim of preparing a series of fluorescent metallacycles with the tunable fluorescence properties, nine 120° triarylamine-based dipyridyl donor ligands **L1–L9** with different pendant functional groups para to the tertiary amine core were designed and synthesized (Fig. 1 and Supplementary Fig. 189–197). Different pendant functional groups, such as $-CF_3$, $-NO_2$, $-CH = C(CN)_2$, $-CHO$, $-H$, $-CH_3$, $-OCH_3$, $-NH_2$, and $-N(CH_3)_2$ were selected to control the push–pull electronic effect of the substituents, thereby switching the PET and ICT properties of the building blocks. All ligands **L1–L9** were well characterized by using multinuclear NMR {$^1H$ and $^{13}C$} spectroscopy and HR-MS (EI) spectrometry (Supplementary Fig. 4–23 and 84–93). Moreover, the structures of ligands **L3**, **L4**, **L6**, and **L8** were unambiguously confirmed by means of single-crystal X-ray diffraction, which revealed that the angle between the two pyridyl rings was close to 120° (Supplementary Fig. 185 and Supplementary Tables 1–4).

According to the design principles of coordination-driven self-assembly, the combination of three 120° donor ligands and three 120° acceptor ligands can result in the formation of a discrete hexagonal metallacycle. Stirring the mixtures of the 120° triarylamine-based dipyridyl donor ligands **L1–L7**, and **L9** with the 120° di-Pt(II) acceptor **8** in a 1:1 ratio in acetone at 40 °C for 4 h led to the formation of discrete hexagonal metallacycles **H1–H7**, and **H9**, respectively (Fig. 1 and Supplementary Fig. 198–206). It should be noted that the preparation of **H8** was conducted through the reaction of ligands **L8** and **8** in methanol at 60 °C for 4 h.

Multinuclear NMR ($^1H$, $^{13}C$, $^{31}P$, 2-D DOSY, $^1H-^1H$ COSY, and NOESY) analysis of the assemblies **H1–H9** revealed the formation of discrete metallacycles with highly symmetric hexagonal scaffold (Supplementary Fig. 24–77). For instance, in the $^1H$ NMR spectrum of each assembly, the α-hydrogen and β-hydrogen nuclei of the pyridine rings exhibited downfield shifts because of the loss of electron density that occurred upon coordination of the pyridine-N atom with the Pt(II) metal center. As shown in the $^1H$ NMR spectrum of metallacycle **H4**, the α- and β-pyridyl hydrogen signals both exhibited the obvious downfield shifts (for α-H, $\Delta\delta \approx 0.04$ ppm, for β-H, $\Delta\delta \approx 0.45$ ppm), which were attributed to the formation of platinum−nitrogen (Pt−N) bonds (Fig. 2a). All $^{31}P\{^1H\}$ NMR spectra of metallacycles **H1–H9** displayed a sharp singlet (for example, ca. 13.73 ppm for **H4** shifted upfield from the precursor **8** by ∼ 6.05 ppm, Fig. 2b). This change, as well as the decrease in the coupling of the flanking $^{195}Pt$ satellites, was consistent with back-donation from the platinum atoms. The sharp NMR signals in both the $^{31}P$ and $^1H$ NMR spectra of metallacycles **H1–H9** along with the good solubility ruled out the formation of oligomers for each assembly. Moreover, for all nine metallacycles **H1–H9**, only one set of

signals was observed in 2-D DOSY spectra for each metallacycle, thus indicating the existence of the sole species. Further characterization of all metallacycles **H1–H9** by $^1H-^1H$ COSY and NOESY exhibited obvious cross peaks between the signals of the pyridine protons (α-H and β-H) and the $PEt_3$ protons ($-CH_2$ and $-CH_3$), which showed good agreement with the formation of the discrete metallacycles **H1–H9** based on the formation of Pt−N bonds.

Investigation of electrospray ionization time-of-flight mass spectrometry (ESI-TOF-MS) provided further support for the formation of discrete hexagonal metallacycles **H1–H9** (Supplementary Fig. 94–102). For instance, as shown in Fig. 2c, the mass spectrum of metallacycle **H4** displayed two main peaks at $m/z = 1177.06$ and $m/z = 911.91$, corresponding to different charge states resulted from the loss of trifluoromethanesulfonate counterions $[M−4OTf^−]^{4+}$ and $[M−5OTf^−]^{5+}$, respectively, where M represents the intact assembly. The isotopic resolution of each peak agreed well with the theoretical distribution, which allowed for the molecularity of the hexagonal metallacycle to be unambiguously established.

All attempts to grow X-ray-quality single crystals of these hexagonal metallacycles **H1–H9** have so far proven unsuccessful. Therefore, the PM6 semiempirical molecular orbital method was utilized to acquire further insight into the structural characteristics of these hexagonal metallacycles. Molecular simulation indicated that all hexagonal metallacycles **H1–H9** exhibited a very similar and roughly planar hexagonal ring at their core surrounded by three triarylamine units with an internal diameter of ∼ 3.36 nm (Supplementary Fig. 158–162).

**Photophysical investigations.** The photophysical properties of metallacycles **H1–H9** were then investigated. As shown in Fig. 3a and Supplementary Fig. 105–123, most of metallacycles exhibited two sharp absorption bands or a main band companied with a shoulder peak centered at ∼ 375 nm. The high-energy absorption bands were almost unaffected by the nature of pendant functional groups para to the tertiary amine core. However, unlike the high-energy absorption bands, the low-energy absorption bands of metallacycles **H1–H9** were very sensitive to the nature of pendant functional groups para to the tertiary amine core, which endows the maximal absorption wavelengths of metallacycles **H1–H9** ranging from 390 nm to 450 nm. Although the similar phenomenon was observed in the absorption spectra of ligands **L1–L9**, most of metallacycles **H1–H9** displayed appreciable red-shifts in the absorption spectra when compared with their corresponding ligands **L1–L9**. For instance, the lowest-energy absorption band for ligand **L5** was centered at 361 nm, whereas the metallacycle **H5** constructed from this ligand **L5** had a band centered at 414 nm. The observed results were attributed to the coordination of metal center with the pyridyl nitrogen, which enriched the ligand π-system, lowered the energy required for excitation, and helped to stabilize the LUMO[43].

The fluorescence emission spectra of ligands **L1–L9** as well as metallacycles **H1–H9** were measured with the results being shown in Fig. 3b and Supplementary Fig. 105–123. The emission spectra for both ligands **L1–L9** and metallacycles **H1–H9** exhibited single bands with the maximum emission wavelength of metallacycles **H1–H9** ranging from 480 nm to 590 nm. In addition, it was worth noting that the color of the fluorescence of all ligands was markedly changed upon the formation of metallacycles (Fig. 3c, f). For instance, the solution of ligand **L1** emitted blue fluorescence while metallacycle **H1** displayed green fluorescence in the same solvent. Interestingly, the metallacycles **H1–H4** with electron-withdrawing groups para to the tertiary amine core displayed the higher emission intensity compared

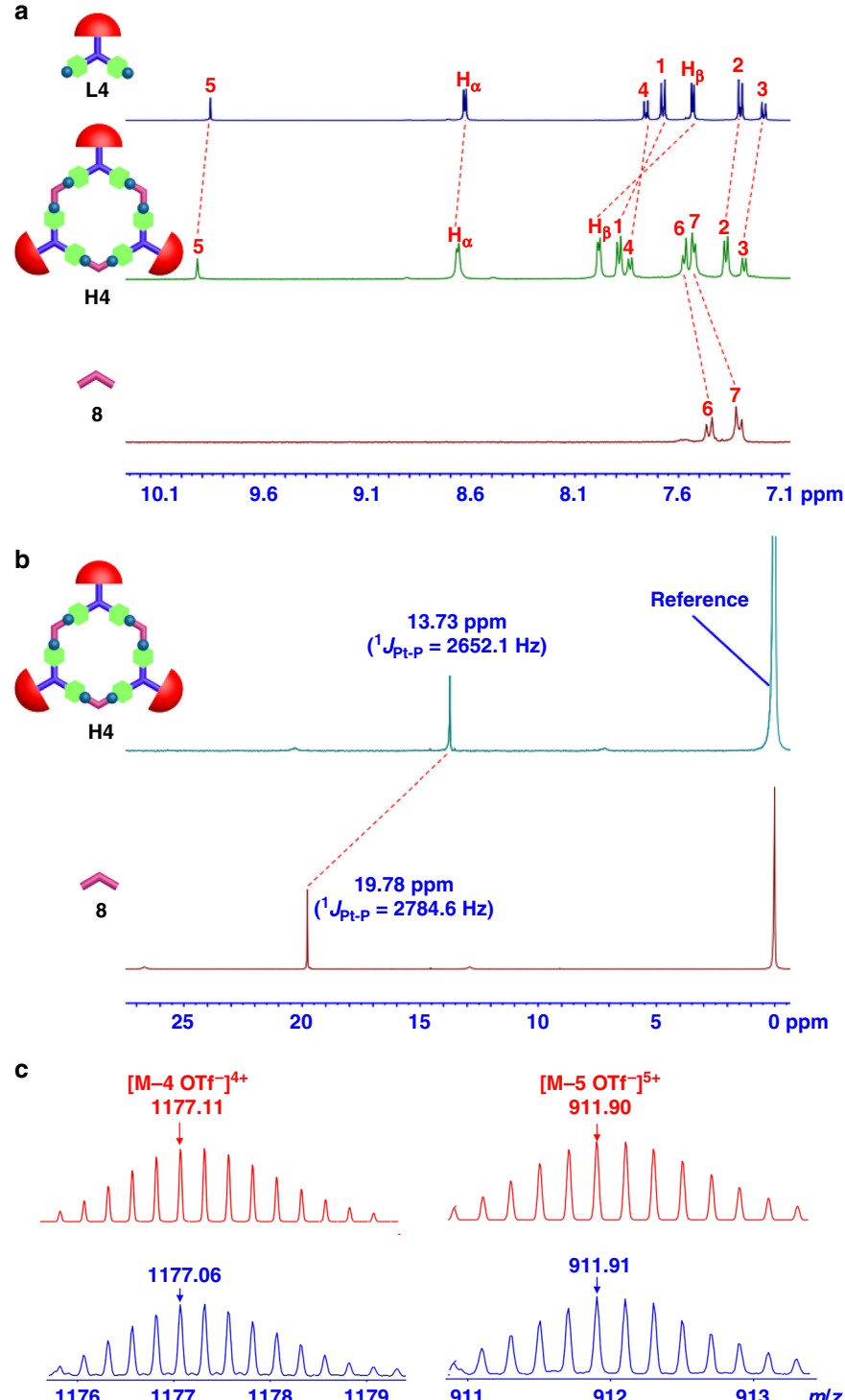

**Fig. 2** NMR spectra and ESI-TOF-MS spectra of **L4** and **H4**. **a** Partial $^1$H NMR spectra (500 MHz, in CD$_2$Cl$_2$) of ligand **L4** (top), 120° dipyridine donor **8** (bottom), and metallacycle **H4** (middle). **b** $^{31}$P NMR spectra (202 MHz, in CD$_2$Cl$_2$) of metallacycle **H4** (top) and 120° dipyridine donor **8** (bottom). **c** Theoretical (top) and experimental (bottom) ESI-TOF-MS spectra of metallacycle **H4**

with their constituent ligands **L1**–**L4** (3.0 equivalents of ligand relative to each corresponding metallacycle). However, unlike **H1**–**H4**, the metallacycles **H5**–**H9**, which contained electron-donating groups para to the tertiary amine core, exhibited the lower emission intensity compared with their constituent ligands **L5**–**L9**. In order to obtain a deeper insight into the photophysical properties of ligands **L1**–**L9** and metallacycles **H1**–**H9**, three-dimensional excitation−emission matrix (3-D EEMs) technology

was employed to investigate the emission spectra. As shown in Supplementary Fig. 186, the samples mappings were collected in all ligands **L1**–**L9** and metallacycles **H1**–**H9**. It could be found that the formation of metallacycles resulted in an apparent change in the EEM peaks. For example, the formation of metallacycle **H1** led to a red shift in the peak wavelengths ($E_x/E_m$) from 341/420 nm to 400/518 nm. However, as the formation of metallacycle **H2**, the fluorescence maximum was red-shifted from

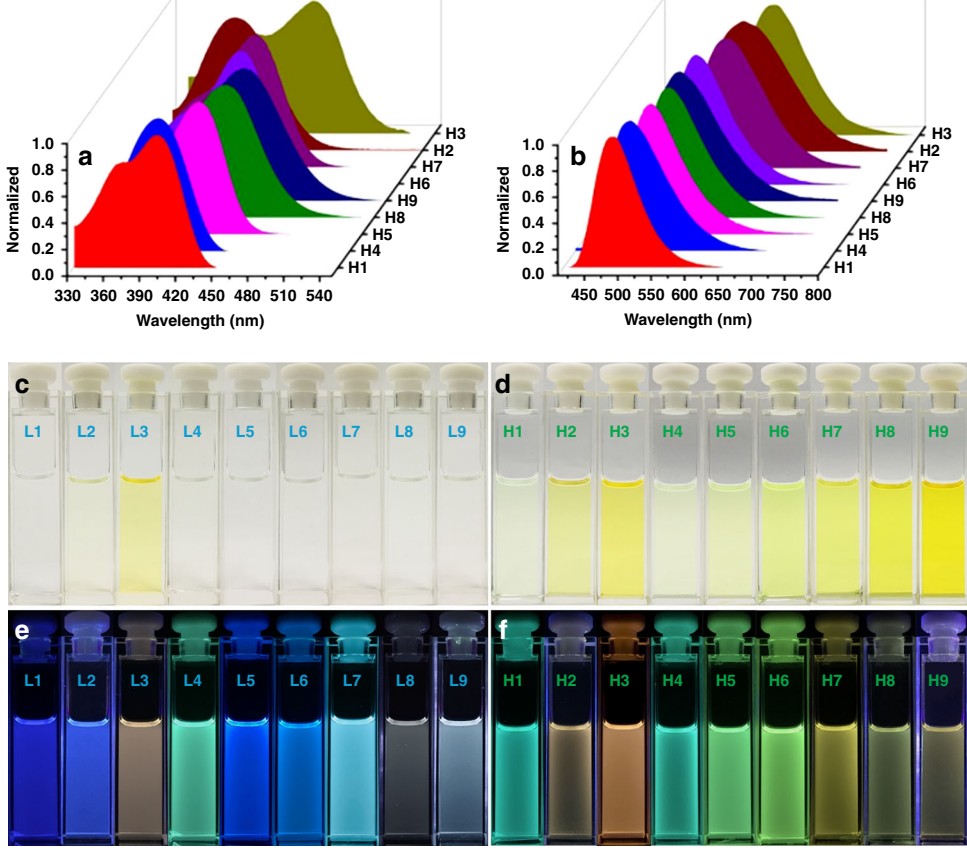

**Fig. 3** Absorption, emission spectra, and photographs of **L1–L9** and **H1–H9**. Normalized absorption **a** and fluorescence emission **b** spectra of metallacycles **H1–H9** in dichloromethane. Photographs of ligands **L1–L9** in dichloromethane in visible light **c** and under 365 nm light excitation **e**. Photographs of metallacycles **H1–H9** in dichloromethane in visible light **d** and under 365 nm light excitation **f**

| Table 1 Fluorescence quantum yields of ligands L1–L9 and metallacycles H1–H9 in DCM (5 μM) | | | | |
|---|---|---|---|---|
| Substituent | Ligand | Fluorescence quantum yield (%) | Metallacycle | Fluorescence quantum yield (%) |
| $-CF_3$ | **L1** | 56 | **H1** | 61 |
| $-NO_2$ | **L2** | 2.1 | **H2** | 0.6 |
| $-CH=C(CN)_2$ | **L3** | 13 | **H3** | 18 |
| $-CHO$ | **L4** | 41 | **H4** | 53 |
| $-H$ | **L5** | 57 | **H5** | 55 |
| $-CH_3$ | **L6** | 60 | **H6** | 52 |
| $-OCH_3$ | **L7** | 55 | **H7** | 8.8 |
| $-NH_2$ | **L8** | 2.5 | **H8** | 2.0 |
| $-N(CH_3)_2$ | **L9** | 1.1 | **H9** | 0.5 |

$E_x/E_m$ 354/455 nm to $E_x/E_m$ 412/572 nm with the difference in $E_x/E_m$ wavelengths as much as 58/117 nm. Different $E_x/E_m$ wavelength shifts produced by the formation of different metallacycle suggested that the tuning of the emission wavelengths could be achieved. Moreover, 3-D EEMs measurements of ligands **L1–L9** and metallacycles **H1–H9** were consistent with the 1-D fluorescence results and no other light-emitting species was observed, which demonstrated that the metallacycles **H1–H9** showed the independence of emission from excitation.

As mentioned above, generally, the fluorescence quantum yields of platinum(II)-pyridyl SCCs are lower than their precursor ligands mainly owing to the well-known heavy-atom effect of platinum. The fluorescence quantum yields of the ligands **L1–L9** and metallacycles **H1–H9** were determined as shown in Table 1. Unlike the previously reported platinum(II)-pyridyl SCCs whose fluorescence quantum yields were usually lower than 20% even if

the fluorescence quantum yields of their constituent ligands exceed 50%, metallacycles **H1**, **H4**, **H5**, and **H6** showed high fluorescence quantum yields of 61%, 53%, 55%, and 52%, respectively. More importantly, as expected, upon the formation of hexagonal metallacycles, the enhancement of fluorescence quantum yields was observed for metallacycles **H1**, **H3**, and **H4** containing electron-withdrawing groups para to the tertiary amine core because the PET process from pyridine to fluorophore was blocked by coordination of the pyridines to the platinum atoms. Moreover, in these PET systems, as pyridyls make little contribution to the fluorescence emission, the heavy-atom effect arising from the platinum ion could be avoided even under the formation of platinum(II)-pyridyl SCCs. Although metallacycle **H2** contains electron-withdrawing group (nitro group) para to the tertiary amine core, metallacycle **H2** displayed an extreme low fluorescence quantum yield (~1%) similar to that of its

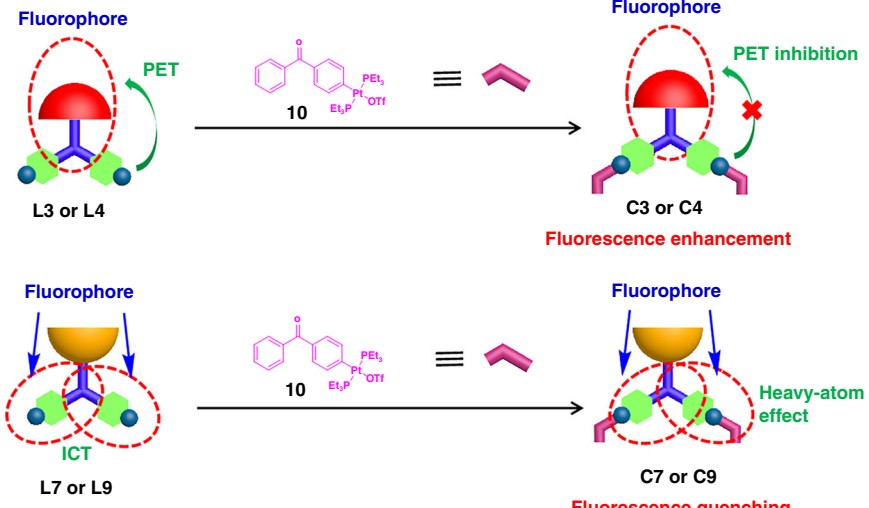

**Fig. 4** Cartoon illustration of the mechanisms of fluorescence changes of ligands **L3**, **L4**, **L7**, and **L9** responding to mono-Pt(II) ligand **10**

constituent ligand **L2** (~ 2%). This observation was attributed to the presence of nitro-groups in the triarylamine skeleton of **H2** and **L2** that caused the usual decrease of radiative deactivation in the singlet state in favor of significant ISC to the triplet state, thus quenching the fluorescence. The similar nitro group effect has been previously reported in literatures[55]. For the metallacycles **H5**–**H9** containing electron-donating groups, para to the tertiary amine core, the fluorescence quantum yields of them were determined to be lower than those of their free ligands, which was consistent with a Pt-enhanced ISC rate as well as the enhanced ICT owing to the presence of the heavy metal ions. Therefore, these results suggested that the emission wavelengths of metallacycles **H1**–**H9** were largely depended upon their substituents, which induced the change of emission colors of metallacycles **H1**–**H9** from blue to green to orange by simply varying the functional groups para to the tertiary amine core. Moreover, the fluorescence quantum yields of metallacycles **H1**–**H9** could be also finely tuned up to 61%. More importantly, by modulating the electron-withdrawing groups or electron-donating groups, the decrease or even increase in the fluorescence quantum yields of metallacycles **H1**–**H9** could be regulated as well when compared with their constituent ligands.

**Mechanism studies**. In order to ensure that such changes in the emission wavelengths and intensities were caused by the platinum (II)-pyridyl coordination, the control experiment was conducted by adding mono-Pt(II) ligand **10** into the solution of ligands **L1**–**L9**, and meanwhile, the monitoring of the changes in both absorption and fluorescence spectra were accomplished. As mono-Pt(II) ligand **10** only contains one platinum, it cannot self-assemble with ligands **L1**–**L9** to form a series of hexagonal metallacycles but can coordinate with them to generate $[2+1]$ model complexes (Fig. 4). Ligands **L3**, **L4**, **L7**, and **L9** were selected as the representative ligands since two of them contain electron-withdrawing groups para to the tertiary amine core and the other two ligands contain electron-donating groups para to the tertiary amine core. The fast formation of $[2+1]$ model complexes **C3**, **C4**, **C7**, and **C9** through the reaction of ligands **L3**, **L4**, **L7**, and **L9** with mono-Pt(II) ligand **10** was confirmed by multinuclear NMR ([1]H and [31]P) titration experiments (Supplementary Fig. 125–132) and ESI-TOF-MS spectra (Supplementary Fig. 133–136). Thus, the absorption and fluorescence responses of ligands **L3**, **L4**, **L7**, and **L9** to mono-Pt(II) ligand **10** were investigated. As shown in Fig. 5, the gradual addition of 2.0

equivalents of mono-Pt(II) ligand **10** into the solution of ligand **L3** or **L4** induced the marked increase of fluorescence intensity. The fluorescence enhancement for the formation of $[2+1]$ model complex can be explained by the inhibition of PET process from pyridines to luminescent moiety. However, an obvious red shift in the absorption wavelength and a decrease in the emission intensity were observed when 2.0 equivalents of mono-Pt(II) ligand **10** was gradually added into the solution of ligand **L7** or **L9**, respectively. The red shift of wavelengths was attributed to the enhancement of push–pull electronic properties of luminescent molecule. More importantly, the changes in the emission wavelengths and intensities were consistent with those of the formation of metallacycles, which suggested that the changes in photophysical properties were achieved through the coordination of platinum(II) and pyridyl. Furthermore, the absorption and fluorescence responses of ligands **L3**, **L4**, **L7**, and **L9** to mono-Pt (II) ligand **10** were in accord with classical PET and ICT mechanisms, respectively, which gave further support for our proposed mechanisms.

With the aim to gaining more insight into the mechanisms for the changing of the fluorescence quantum yields of metallacycles **H1**–**H9** upon the formation of metallacycles, time-resolved intensity decays of both ligands **L1**–**L9** and metallacycles **H1**–**H9** were measured by time-correlated single-photon counting (TCSPC) method (Supplementary Fig. 137–138 and Supplementary Tables 7–8). The fluorescence lifetime measurements of ligands **L1**–**L9** and metallacycles **H1**–**H9** revealed that the fluorescence intensity decays were multiexponential and all of them were within nanosecond scale, thus suggesting a typical fluorescent emission. For instance, the decay of metallacycle **H1** was found to follow a biexponential function with a shorter component of 1.6 ns with a 35% proportion and a longer component of 2.4 ns with a 65% proportion. However, the decay of metallacycle **H3** was found to follow a triexponential function with the components of 0.1 ns, 1.4 ns, and 2.4 ns with the proportions of 22%, 26%, and 52%, respectively. To avoid the possible complexity, an averaged fluorescence lifetime ($\tau$) of each of the ligands **L1**–**L9** and metallacycles **H1**–**H9** was considered rather than emphasizing too much on the individual components. From the average fluorescence lifetimes ($\tau$) of ligands **L1**–**L9** and metallacycles **H1**–**H9** shown in Supplementary Tables 7–8, the radiative rate constant $k_{rad}$ and the nonradiative rate constant $k_{nr}$ were calculated based on following photophysical equations:

$$k_{exc} = 1/\tau \qquad (1)$$

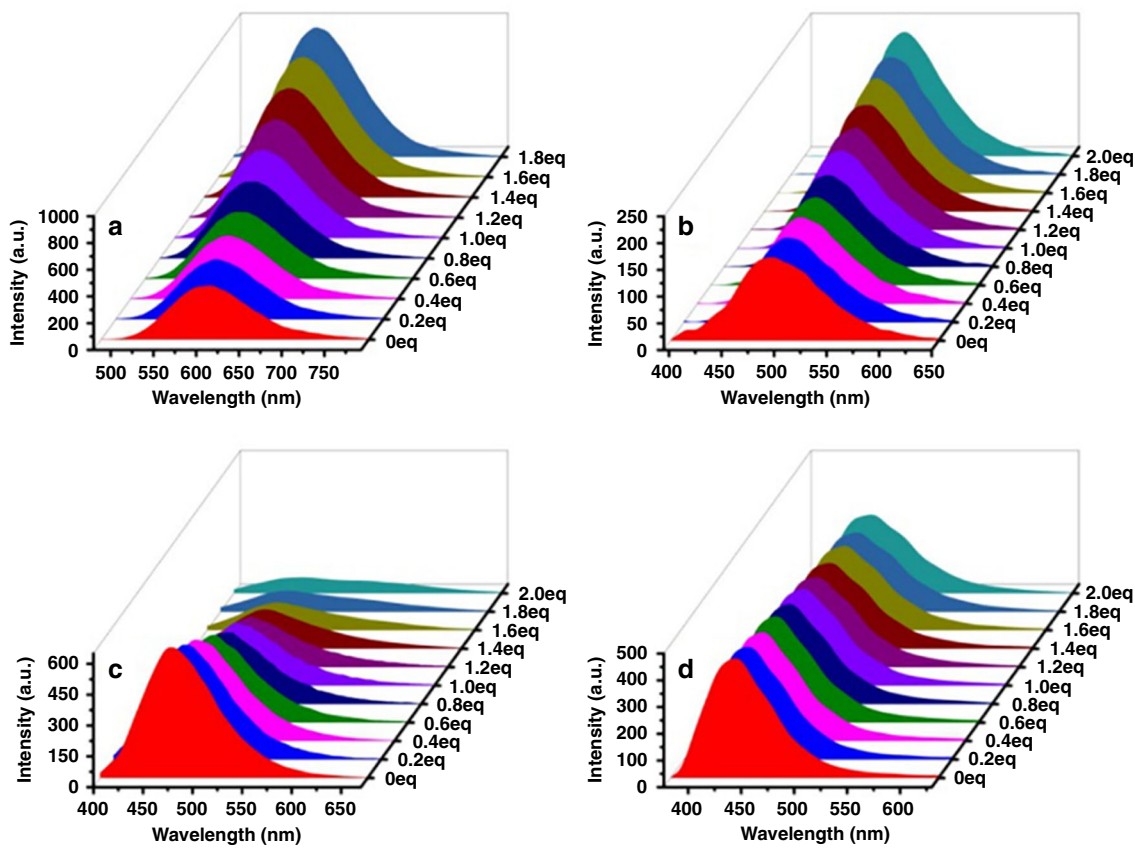

**Fig. 5** Fluorescence titration spectra of ligands **L3**, **L4**, **L7**, and **L9**. Fluorescence spectra of ligands **L3 a**, **L4 b**, **L7 c**, and **L9 d** upon the addition of mono-Pt (II) ligand **10** in dichloromethane

**Table 2 Excitation rate constants ($k_{exc}$), radiative rate constants ($k_{rad}$), and nonradiative rate constants ($k_{nr}$) for ligands L1–L9 and metallacycles H1–H9**

| Substituent | Ligand | $k_{exc}$ (s$^{-1}$) | $k_{rad}$ (s$^{-1}$) | $k_{nr}$ (s$^{-1}$) | Metallacycle | $k_{exc}$ (s$^{-1}$) | $k_{rad}$ (s$^{-1}$) | $k_{nr}$ (s$^{-1}$) |
|---|---|---|---|---|---|---|---|---|
| $-CF_3$ | **L1** | $6.25 \times 10^8$ | $3.48 \times 10^8$ | $2.77 \times 10^8$ | **H1** | $4.55 \times 10^8$ | $2.77 \times 10^8$ | $1.77 \times 10^8$ |
| $-NO_2$ | **L2** | $4.00 \times 10^8$ | $8.56 \times 10^6$ | $3.91 \times 10^8$ | **H2** | $1.67 \times 10^9$ | $1.00 \times 10^7$ | $1.66 \times 10^9$ |
| $-CH=C(CN)_2$ | **L3** | $5.26 \times 10^8$ | $6.83 \times 10^7$ | $4.58 \times 10^8$ | **H3** | $4.76 \times 10^8$ | $8.63 \times 10^7$ | $3.90 \times 10^8$ |
| $-CHO$ | **L4** | $1.61 \times 10^8$ | $6.66 \times 10^7$ | $9.47 \times 10^7$ | **H4** | $5.00 \times 10^8$ | $2.64 \times 10^8$ | $2.36 \times 10^8$ |
| $-H$ | **L5** | $4.55 \times 10^8$ | $2.61 \times 10^8$ | $1.94 \times 10^8$ | **H5** | $3.85 \times 10^8$ | $2.10 \times 10^8$ | $1.75 \times 10^8$ |
| $-CH_3$ | **L6** | $3.85 \times 10^8$ | $2.30 \times 10^8$ | $1.55 \times 10^8$ | **H6** | $3.23 \times 10^8$ | $1.69 \times 10^8$ | $1.54 \times 10^8$ |
| $-OCH_3$ | **L7** | $2.56 \times 10^8$ | $1.40 \times 10^8$ | $1.16 \times 10^8$ | **H7** | $1.25 \times 10^9$ | $1.10 \times 10^8$ | $1.14 \times 10^9$ |
| $-NH_2$ | **L8** | $5.56 \times 10^8$ | $1.41 \times 10^7$ | $5.42 \times 10^8$ | **H8** | $4.00 \times 10^8$ | $7.80 \times 10^6$ | $3.92 \times 10^8$ |
| $-N(CH_3)_2$ | **L9** | $3.85 \times 10^8$ | $4.27 \times 10^6$ | $3.80 \times 10^8$ | **H9** | $8.33 \times 10^8$ | $4.25 \times 10^6$ | $8.29 \times 10^8$ |

$$k_{rad} = \Phi_F * k_{exc} \tag{2}$$

$$k_{exc} = k_{rad} + k_{nr} \tag{3}$$

where $k_{exc}$ was excitation rate constant, $k_{rad}$ and $k_{nr}$ represented the radiative rate constant and nonradiative rate constant, respectively. For each ligand or metallacycle, $k_{exc}$ could be calculated from Eq. (1), and $k_{rad}$ could be measured from Eq. (2) using the experimentally determined fluorescence quantum yield ($\Phi_F$) value. Finally, once the $k_{exc}$ and $k_{rad}$ were known, the $k_{nr}$ could be deduced from Eq. 3. As shown in Table 2, for example, although the nonradiative rate constants ($k_{nr}$) of metallacycles **H5–H8** were not larger than those of their corresponding ligands, the value of the ratio nonradiative rate constant ($k_{nr}$)/radiative rate constant ($k_{rad}$) of each metallacycle (**H5–H8**) was larger than

that of the corresponding ligand, which provided a quantitative understanding of the lower quantum yields of metallacycles **H5–H8** compared with their constituent ligands **L5–L8**. Therefore, the obtained $k_{rad}$ and $k_{nr}$ afforded quantitative insights into the reasons for the changing of the fluorescence quantum yields of metallacycles **H1–H9** upon the formation of metallacycles.

With the aim of confirming whether the emission properties of metallacycles **H5–H9** is fluorescence or phosphorescence, the delayed emission spectra of representative metallacycles **H5–H9** were measured. As shown in Supplementary Fig. 141, very weak emission from metallacycles **H5–H7** and almost no emission from metallacycles **H8–H9** were observed. Moreover, the normalized steady-state emission spectra of metallacycles **H5–H7** were coincident with those of their delayed emission spectra (Supplementary Fig. 142). Furthermore, the emission lifetime measurements of metallacycles **H5–H9** revealed that all of them

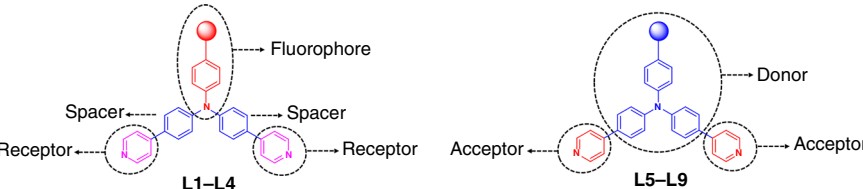

**Fig. 6** Schematic illustrations of PET/ICT of ligands **L1**-**L9**

were within nanosecond scale. The results further indicated that the emission properties of metallacycles **H5**–**H9** is fluorescence rather than phosphorescence.

In order to further elucidate the mechanism and the effect of substituted groups in the tuning of fluorescence emission wavelength and fluorescence quantum yield, molecular simulation by using time-dependent density functional theory (TD-DFT) were carried out for both ligands **L1**–**L9** and metallacycles **H1**–**H9**. As shown in Supplementary Fig. 163–180, for ligands **L2**–**L4**, which contained electron-withdrawing groups para to the tertiary amine core, their emission was mainly generated from the part of aniline and electron-withdrawing groups as witness by most of the electron densities of HOMO and LUMO were localized on the part of aniline and electron-withdrawing groups. Notably, ligand **L1** and metallacycle **H1** were exceptions, in which most of the electron densities were localized on the part of aniline and pyridine units although they were also substituted by electron-withdrawing group (trifluoromethyl), which might be caused by the well-known "fluorine effect". In these ligands **L2**–**L4**, the part of aniline and electron-withdrawing groups acted as a fluorophore, pyridine units served as a receptor, and benzene units were the spacer that linked the two parts of fluorophore and receptor (Fig. 6). It indicated that they were PET molecules. Without binding of platinum(II), PET process occurred owing to the transfer of electron density, originating at the lone pair electrons on N atoms of the pyridine units, to the LUMO of the fluorophore. However, binding of platinum(II) to pyridine during the formation of metallacycles **H3**–**H4** hindered the PET process, thus leading to the enhancement of fluorescence intensity and quantum yield. Although the molecular simulation results also demonstrated that the binding of platinum(II) to pyridine during the formation of metallacycle **H2** hindered the PET process, both nitro group-containing metallacycle **H2** and ligand **L2** displayed extremely low quantum yields owing to the strong S1 → T1 ISC caused by nitro group. It should be noted that pyridine units in metallacycles **H2**–**H4** just served as a receptor rather than a fluorophore in this case. Therefore, the heavy-atom effect arising from the platinum ion could be avoided even under the formation of platinum(II)-pyridyl SCCs.

For ligands **L5**–**L9**, which contained electron-donating groups para to the tertiary amine core, their emission was mainly generated from the part of aniline and pyridine units as the electron densities of HOMO of ligands **L5**–**L9** were distributed over the whole molecule while most of the electron densities of LUMO were localized on the part of aniline and pyridine units. In these ligands **L5**–**L9**, the part of aniline and pyridine units acted as a fluorophore, in which aniline and pyridine units served as donor and acceptor, respectively (Fig. 6). It means that ligands **L5**–**L9** were ICT molecules. Electron-donating groups such as −H, −CH₃, −OCH₃, −NH₂, and −N(CH₃)₂ could regulate the electron-donating ability of aniline, and then changed the emission wavelength of these fluorescent ligands. When the electron-accepting part pyridine units interacted with platinum(II), the electron-accepting character of the fluorescent molecule increases, thus generating a red shift in the emission spectrum. In the ligands **L5**–**L9**, pyridine units were important

part of fluorophore and made large contribution to the fluorescence emission. The coordination of platinum(II) with pyridine during the formation of metallacycles **H5**–**H9** could induce the decrease of fluorescence intensity and fluorescence quantum yield due to the heavy-atom effect arising from the platinum ion.

In order to provide the more convincing molecular calculations, ligands **L3** and **L7** as well as metallacycles **H3** and **H7** were selected as the representatives to investigate the whole molecular simulation through the analysis of hole and electron distribution by using TD-DFT. As shown in Supplementary Fig. 181, for ligand **L3**, holes and electrons were mainly located at the part of aniline and electron-withdrawing groups, which indicated that electrons transferred from the aniline groups to the electron-withdrawing groups when ligand **L3** was excited to the S1 state. However, for ligand **L7**, holes and electrons were distributed at the part of aniline and pyridine units (Supplementary Fig. 182). This result showed that electrons transferred from the aniline groups to the pyridine units when ligand **L7** was excited to the S1 state. For metallacycle **H3**, the hole-electron distributions of S1, S2, S3 states, which derived from three ligands respectively, were similar with each other (Supplementary Fig. 183). More importantly, all of them were similar with the hole-electron distributions of S1 states of the corresponding ligand **L3**. These results suggested that electrons also transferred from the aniline groups to the pyridine units when excitation of metallacycle **H3**. Similar phenomena were also observed in metallacycle **H7** (Supplementary Fig. 184).

More-detailed investigations of TD-DFT calculations were conducted by selecting ligand **L3**, metallacycle **H3**, ligand **L7**, and metallacycle **H7** as representatives since they contain electron-withdrawing groups and electron-donating groups para to the tertiary amine core, respectively. The results were shown in Fig. 7 and Supplementary Table 11. In each case, the long wavelength absorption and emission were assigned to a transition between S0 ↔ S1 (S0: ground-state, S1: singlet excited state), mainly corresponding to the promotion of an electron from HOMO to LUMO on the chromophore core. They were characteristic π → π* absorption and emission from the chromophore core. Moreover, the orbital transition from HOMO to LUMO + 1 was also observed in ligand **L3**, metallacycle **H3**, ligand **L7**, and metallacycle **H7**. For ligand **L3** and metallacycle **H3**, although molecular orbital LUMO exhibited electron density centered on the part of aniline and electron-withdrawing groups, molecular orbital LUMO + 1 displayed electron density mainly centered on the part of aniline and pyridine units. Supplementary Table 11 clearly illustrated that the oscillator strength (f, which is a very important parameter to evaluate the possibility of a transition) of the transition between HOMO and LUMO was bigger than that of the transition between HOMO and LUMO + 1 in both ligand **L3** and metallacycle **H3**, which indicated the emission of ligand **L3** and metallacycle **H3** was mainly generated from the part of aniline and electron-withdrawing groups and PET played more important role in tuning of the fluorescence emission wavelength and fluorescence quantum yield of ligand **L3** and metallacycle **H3**. However, for ligand **L7** and metallacycle **H7**, both molecular

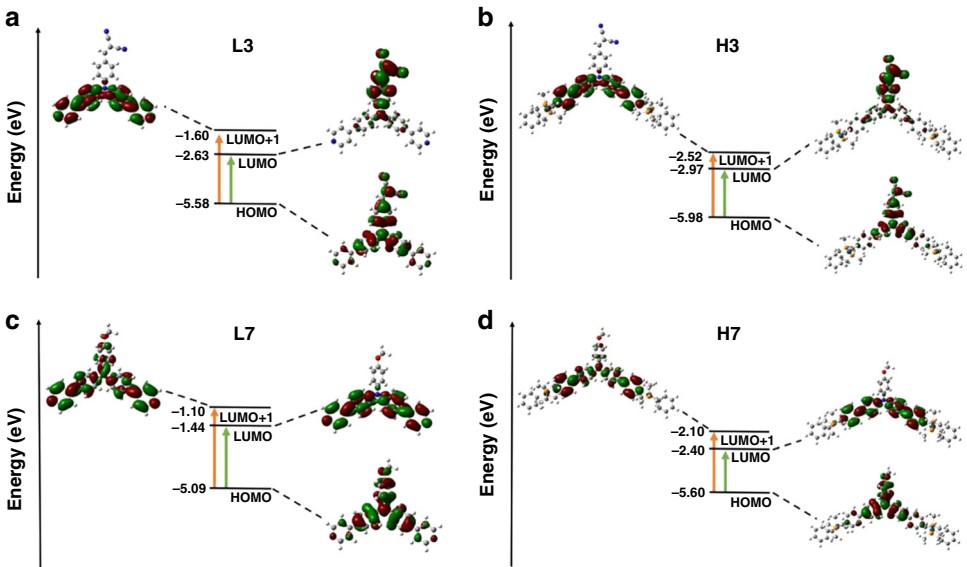

**Fig. 7** TD-DFT transitions for **L3**, **L7**, **H3**, and **H7**. Predicted TD-DFT transitions for ligands **L3 a**, **L7 c**, and metallacycles **H3 b**, **H7 d** model with oscillator strengths above 0.3

orbitals LUMO and LUMO + 1 were mainly localized on the part of aniline and pyridine units, which suggested that the ICT emission from aniline to pyridine units was the main factor in tuning of the fluorescence emission wavelength and fluorescence quantum yield of ligand **L7** and metallacycle **H7**. These results further supported the mechanisms proposed above.

Moreover, in order to ensure that the enhancement of fluorescence for metallacycles **H1**, **H3**, and **H4** were caused by the inhibition of PET process through the platinum(II)-pyridyl coordination, a representative control experiment was conducted by mixing dipyridyl ligand **L3** and platinum−bromine complex **11** for 6.0 h (Supplementary Fig. 187). The platinum atoms in complex **11** were protected by bromine moieties, which cannot coordinate with ligand **L3** to form a metallacycle. As shown in Supplementary Fig. 188, almost no change in the absorption and emission spectra of ligand **L3** was observed, which further demonstrated that both reasons for inhibiting PET process and the driving force for metallacycle formation were the platinum (II)-pyridyl coordination.

**The preparation of versatile fluorescent materials**. By taking advantage of their excellent fluorescence properties, including tunable fluorescence wavelengths, tunable fluorescence quantum yields, and high emission, metallacycles **H1**–**H9** were applied to prepare versatile fluorescent materials. The construction of dye-loaded thin-layer chromatography (TLC) plates has been attracted increasing interesting, as their strong mechanical strength, high portability, simple operability, and broad applications in the fluorescence sensing of explosive gases and volatile gases[56,57]. So, a series of fluorescent metallacycles **H1**–**H9**-loaded TLC plates were manufactured simply in the following way. Nine TLC plates were immersed into the dichloromethane solution of metallacycles **H1**–**H9**, respectively, and they were then dried in the air for several hours. As displayed in Fig. 8a, the prepared **H1**–**H9**-loaded plates emitted different color fluorescence upon excitation under a hand-held UV lamp with excitation wavelength at 365 nm. For instance, the metallacycle **H3**-loaded plate emitted an orange fluorescence while the metallacycle **H7**-loaded plate emitted a green fluorescence.

Despite many organic materials exhibited high emission efficiency in solution, lots of them were non-emissive in the

solid state because of the fluorescence quenching caused by intermolecular interactions occurring in the condensed phase[58]. Herein, metallacycles **H1**, **H3**, and **H7** were selected as the representatives to investigate the fluorescence properties in the solid state through the preparation of fluorescent polymer films. It is known that the fluorescent polymer films feature some prominent advantages, such as reusability and easy fabrication into devices[59,60]. **H1**, **H3**, and **H7**-functionalized polymer films were obtained by dropping the DMF solution of metallacycles **H1**, **H3**, and **H7** (0.1 wt%) doped with polyvinylidene fluoride (PVDF) onto the $SiO_2$ substrate surface at ambient temperature, respectively, and then kept them at 70 °C in the oven for 6.0 h to evaporate the solvent. The absorption and fluorescence emission spectra of these three films in the dry state were shown in Supplementary Fig. 144–146. **H1**-functionalized polymer film ($\lambda_{em}$ = 508 nm) displayed blue-shifted emission when being compared with metallacycle **H1** in solution ($\lambda_{em}$ = 487 nm), however, both **H3** and **H7**-functionalized polymer films ($\lambda_{em}$ = 576 nm for **H3**, $\lambda_{em}$ = 516 nm for **H7**) exhibited an obvious blue-shifted emission when compared with metallacycles **H3** and **H7** in solution ($\lambda_{em}$ = 586 nm for **H3**, $\lambda_{em}$ = 555 nm for **H7**). It might be attributed to the different aggregation behaviors of metallacycles **H1**, **H3**, and **H7** within the film. Notably, **H1**, **H3**, and **H7**-functionalized polymer films emitted blue fluorescence, orange fluorescence, and green fluorescence with high fluorescence quantum yield up to 29%, 18%, and 23%, respectively. Moreover, by taking advantage of the easy processing and good elasticity of polymer film, 3-D fish-shape fluorescent films exhibiting bright fluorescence with different colors were fabricated by hand (Fig. 8b–g). Moreover, fluorescent films fabricated from metallacycles and ligands exhibited different fluorescence properties (Supplementary Fig. 148). For example, 3-D fish-shape fluorescent films prepared from ligands **L1** displayed green fluorescence while the fluorescent films made from metallacycle **H1** emitted blue fluorescence with the stronger emission.

Inkjet printing, as a cheap, easily handled yet powerful technique for creating highly-defined patterns, has been used for conductive circuits, flexible electronics, and fingerprint recognition[61–63]. So far, single-walled carbon nanotubes, luminescent CdTe nanocrystals, carbon dots, and fluorescein have been patterned using this method[64,65]. However, to the best of

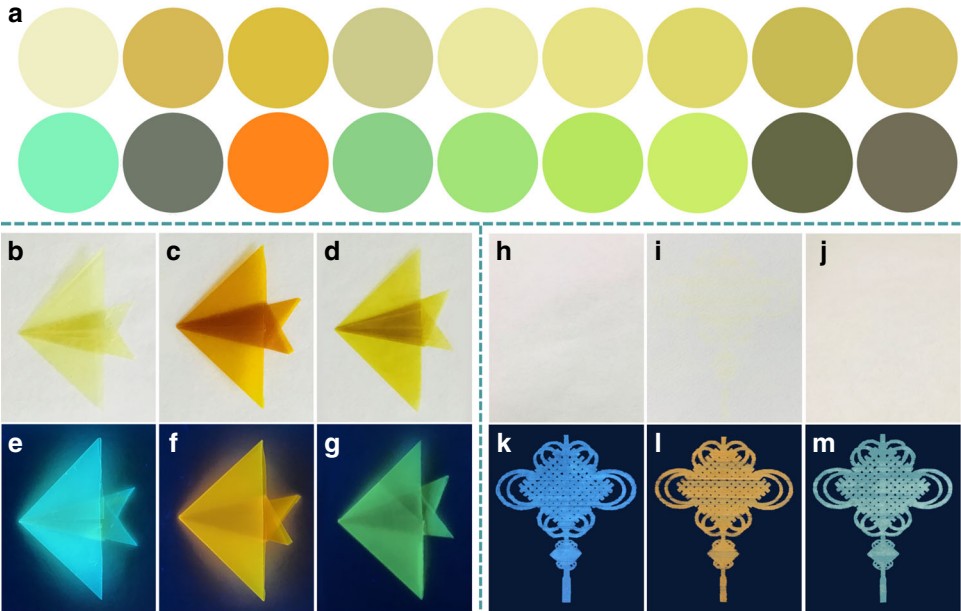

**Fig. 8** Versatile fluorescent materials. **a** Visual (top) and fluorescence colors (bottom) of metallacycles **H1**–**H9**-coated TLC plates (Figures showed from left to right were sequentially corresponded to metallacycles **H1**–**H9**). The fluorescence colors were observed by using a hand-held UV lamp with an excitation at 365 nm. Photographs of the flexible fluorescent films **b**–**g** and the fluorescent patterns **h**–**m** by inkjet printing from metallacycle **H1** in visible light **b**, **h** and under 365 nm light excitation **e**, **k**, metallacycle **H3** in visible light **c**, **i** and under 365 nm light excitation **f**, **l**, metallacycle **H7** in visible light **d**, **j** and under 365 nm light excitation **g**, **m**

our knowledge, the use of SCCs for fluorescent inkjet printing has not been reported. Thus, herein, metallacycles **H1**, **H3**, and **H7** were also selected as the representatives to explore the application as fluorescent inks for printing versatile fluorescent patterns. The acetone solution of metallacycles **H1**, **H3**, and **H7** as the inkjet-printing source was printed, respectively. As shown in Fig. 8h–m, all of the patterns printed from metallacycles **H1**, **H3**, and **H7** solutions were invisible under ambient conditions, which resulted in the information of the patterns being unrecognizable by the naked eye. However, it should be noted that three patterns of the Chinese knot showing blue fluorescence, orange fluorescence, and blue-green fluorescence could be observed clearly under the excitation of hand-held UV lamp (365 nm), respectively. The results suggested that all kinds of patterns with multicolor fluorescence and designable shapes could be printed using different types of ink prepared from metallacycles **H1**–**H9**. Moreover, it should be noted that the patterns printed from metallacycles displayed different fluorescence colors compared to the patterns from simple model metal complexes and organic fluorescence ligands (Supplementary Fig. 147–148). This finding further illustrated that PET/ICT regulation strategy was an effective approach to construct organoplatinum metallacycles with tunable fluorescence wavelengths not only in solution but also in solid state. Therefore, the "vis-invisible" and "UV-visible" properties of metallacycles **H1**, **H3**, and **H7** may allow for their potential applications in anti-counterfeiting, information hiding and storage, and optoelectronic devices.

## Discussion

In summary, we presented the example on the construction of organoplatinum metallacycles with high fluorescence quantum yields and tunable fluorescence wavelengths by simply switching the PET and ICT properties of building blocks based on substituent effect. In this study, nine 120° triarylamine-based dipyridyl donor ligands **L1**–**L9** with different pendant functional groups (including $-CF_3$, $-NO_2$, $-CH=C(CN)_2$, $-CHO$, $-H$, $-CH_3$, $-OCH_3$,

$-NH_2$, and $-N(CH_3)_2$) para to the tertiary amine core were designed and synthesized, from which a series of hexagonal organoplatinum metallacycles **H1**–**H9** with different substituents were successfully prepared through coordination-driven self-assembly.

The photophysical properties investigation of metallacycles **H1**–**H9** revealed that metallacycles **H1**–**H4** with electron-withdrawing groups para to the tertiary amine core displayed the higher emission intensity (with the fluorescence quantum yield up to 61%) compared with their constituent ligands **L1**–**L4**, however, the metallacycles **H5**–**H9** containing electron-donating groups para to the tertiary amine core exhibited the lower emission intensity compared with their constituent ligands **L5**–**L9**. Moreover, metallacycles **H1**–**H9** exhibited tunable fluorescence wavelengths ranging from 480 nm to 590 nm and emitted different fluorescence colors in the same solvent. The luminescence mechanisms of these metallacycles **H1**–**H9** were systematically investigated by 3-D EEMs, TCSPC method, TD-DFT calculation, and control experiments as well. The results indicated that the PET and ICT properties of metallacycles could be switched by the modification of substituents, which offered the self-assembled organoplatinum metallacycles with high fluorescence quantum yields and tunable fluorescence wavelengths.

By utilizing the excellent fluorescence properties of resultant metallacycles, many kinds of fluorescent materials including fluorescent metallacycle-loaded TLC plates, 3-D-fish-shape fluorescent films, and fluorescent inks for inkjet printing were successfully prepared. Notably, these metallacycle-functionalized polymer films emitted bright fluorescence with different colors and relatively high fluorescence quantum yield up to 29%. Moreover, three patterns of the Chinese knot with multicolor fluorescence and designable shapes were printed using different types of ink prepared from metallacycles. The "vis-invisible" and "UV-visible" properties of these prepared fluorescent materials offer them wide potential applications in anti-counterfeiting, information hiding and storage, and optoelectronic devices. In conclusion, this study presents the example on the construction of organoplatinum metallacycles with high fluorescence quantum yields and tunable fluorescence wavelengths by

simply switching the PET and ICT properties of building blocks based on the substituent effect.

## Methods

**General information**. All reagents were commercially available and used without further purification. TLC analyses were performed on silica-gel plates, and flash chromatography was conducted using silica-gel column packages. All air-sensitive reactions were carried out under argon atmosphere. $^1$H NMR, $^{13}$C NMR, and $^{31}$P NMR spectra were recorded on a Bruker 300 MHz spectrometer ($^1$H, 300 MHz), a Bruker 400 MHz spectrometer ($^1$H, 400 MHz, $^{13}$C, 100 MHz, $^{31}$P, 161.9 MHz), and a Bruker 500 MHz spectrometer ($^1$H, 500 MHz, $^{13}$C, 126 MHz, $^{31}$P, 202 MHz) at 298 K. The $^1$H and $^{13}$C NMR chemical shifts are reported relative to residual solvent signals, and $^{31}$P NMR resonances are referenced to internal standard sample of 85% $H_3PO_4$ (δ 0.0).

**General procedure for preparation of H1–H9**. The dipyridyl donor ligand **L1–L9** (6.0 μmol) and the organoplatinum 120° acceptor **8** (6.0 μmol) were weighed accurately into a glass vial. To the vial was added 2.0 mL acetone and the reaction solution was then stirred at 40 °C for 4 h to yield a homogeneous solution (for **L8**, 2.0 mL methanol was added into the vial and the reaction solution was stirred at 60 °C for 4 h). Solid product **H1–H9** was obtained by removing the solvent under vacuum.

**UV–Vis absorption, fluorescence emission and excitation spectra**. UV−vis spectra were recorded in a quartz cell (light path 10 mm) on a Cary 50Bio UV −visible spectrophotometer. Steady-state fluorescence spectra and excitation spectra were recorded in a conventional quartz cell (light path 2 mm) on a Cary Eclipse fluorescence spectrophotometer.

**3-D EEM and lifetime experiment**. Three-dimensional excitation−emission matrix spectra were recorded in a quartz cell (light path 2 mm) on a HORIBA FluoroMax-4 fluorescence spectrophotometer.

Lifetime were measured on a home-made system based on the TCSPC technique[66].

**Fluorescence quantum yields**. Fluorescence quantum yields were measured in absolutely in solution using a commercial fluorometer with integrating sphere (RF-6000, shimadzu).

**Fluorescence lifetimes**. The fluorescence lifetimes were measured based on TCSPC technique. The squared deviations obtained by the least-squares analysis was utilized to estimate the fitting of TCSPC data and judge the multiexponential emission decay processes.

**DFT and TD-DFT calculations**. All calculations were performed using the Gaussian09 (G09) program package. The DFT method was employed using the B3LYP hybrid functional with LANL2DZ effective core potentials (Pt) and 6-31 G* (C, H, N, F and O) basis set. Orbitals were visualized using GaussView 5.0 with an isovalue of 0.02. To minimize computational cost, we simplified the compound and replaced it with 1/3 of the compound.

**Geometry optimizations**. The compound **H1–H9** were optimized by the PM6 semiempirical molecular orbital method. And the ligand **L1–L9** were optimized by the DFT method.

## Data availability

The data that support the findings of this study are available from the authors on reasonable request, see author contributions for specific data sets. The X-ray crystallographic coordinates for structures reported in this study have been deposited at the Cambridge Crystallographic Data Centre (CCDC), under deposition numbers 1873682, 1873683, 1873684, and 1873685. These data can be obtained free of charge from The Cambridge Crystallographic Data Centre via www.ccdc.cam.ac.uk/data_request/cif.

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

## Acknowledgements

This work was financially supported by the NSFC/China (Nos. 21625202, 21672070, 21922506, and 21871092), and Shanghai Pujiang Program (No. 18PJD015). We thankfully acknowledge Professor Feng Wang of University of Science and Technology of China for the discussion of fluorescent inkjet printing, Professor Haitao Sun (ECNU) and Mr. Zhubin Hu (ECNU) for the TD-DFT calculations, and Mr. Changxing Zhao of East China University of Science and Technology for the discussion of phosphorescence experiment.

## Author contributions

H.-B.Y., L.X. and J.-L.Z. conceived the project, analyzed the date, and wrote the manuscript. J.-L.Z. performed the most of experiments. J.-L.Z. and Y.-Y.R. completed the synthesis. Y.-Y.R., Y.Z., X.L., G.-Q. Y., B.S., X.C., Z.C., X.-L.Z., H.T., J.C. and X.-P.L. helped in experiments and data analyses. All authors discussed the results and commented on the manuscript.

## Additional information

**Competing interests:** The authors declare no competing interests.

