## [Transparent Peer Review File · Nature Communications]

Reviewers' comments:

Reviewer #1 (Remarks to the Author):

The authors presented a comprehensive study on the tuning of PET/ICT properties through altering the electron donating/accepting ability of the substituent of SCC systems. The fluorescence mechanism of the system has been explored and complexes with high fluorescence quantum yield have been achieved. However, there are problems.

1. Some mistakes have been spotted in the article. For example, in Scheme 2 (page 5), L9 and C9 were mistakenly denoted as L10 and C10 respectively. In Table 1 (page 5) and Table 2 (page 7), –CH₃O should be –OCH₃.

2. The article suggested the luminescence properties of H1-H9 are fluorescence in nature (Page 5). They suggested that in some SCCs, the introduction of platinum center produces fluorescence enhancement due to PET inhibition, but in other SCCs fluorescence quenching due to heavy atom effect that gives significant ISC. The arguments are highly speculative and are not convincing. The authors did not discuss why in some SCCs the fluorescence prefer to be quenched and in others prefer to be enhanced. What are the driving forces for PET in these SCCs before Pt bonding? This can give information on the possibility of PET. There are no supporting evidences for the claims. Can the emission properties of H5-H9 be phosphorescence? Additional experiments should be done to support the assignments.

3. Since phosphorescence can be quenched by oxygen, have the authors tried conducting the photophysical measurements under deoxygenated or oxygenated condition?

4. The quantum yield (Page 5) should be specified as fluorescence quantum yield. They cannot be accurate up to two decimal places in % fluorescence QY. Are the absolute fluorescence QYs measured in solution or solid state? This is not specified. The SCCs are dynamic equilibrium mixtures. How to ensure the fluorescence QYs refer to the SCC or the mixture?

5. Emission quantum yield under degassed media should be conducted to compare with the existing data. This is same for emission decay profiles. The interpretations of multiexponential processes are highly speculative with no supporting evidence.

6. The article demonstrated the utilization of the complexes as fluorescent ink for printing (page 8-9). It is suggested that comments should be added to emphasize the advantages of using metallacycles over simple metal complexes (eg. complex C3, C4) or organic fluorescence ligands (eg. ligand L5, L6) for such purpose.

7. The authors suggested that many organic materials exhibit intense emission only in solution state (page 8). To make the study more comprehensive, it is suggested that some precursor ligands should be selected to investigate their solid-state emission properties (fluorescent films / ink printing).

The novelty of the work is thin and is not up to standard for publication in Nature Comm.

Reviewer #2 (Remarks to the Author):

In this manuscript, the authors report the systematic construction of organoplatinummetallacycles with high quantum yields and tunable fluorescence wavelengths by switching the PET and ICT properties of the building blocks. Detailed controlled experiments and DFT/TDDFT calculations were performed to give a further insight into the photophysical mechanism. Although the reported results are interesting, this manuscript has several issues as listed below:

1. For the comments "the observed results were attributed to the coordination of metal center with the pyridyl nitrogen, which enriched the ligand n-system, lowered the energy required for excitation, and helped to stabilize the LUMO" in the manuscript, a relevant reference should be provided.

2. I feel confused for the fluorescence quantum yield of H2. As shown in Table 1, the quantum yield of L2 is 2.14%, while the quantum yield of H2 is 0.6%. However, the author claim that

“binding of platinum(II) to pyridine during the formation of metallacycles H2-H4 hindered the PET process, thus leading to the enhancement of fluorescence intensity and quantum yield”. Is there a mistake?

3. The authors point out that the coordination of metallacycles H5-H9 can induce the quantum yield due to the heavy atom effect arising from the platinum ion. Accordingly, nonradiative rate constants k_{nr} will be increased. But for the data of H5-H8 as shown in table 2, why is their k_{nr} not larger than that of the corresponding ligands?

4. The authors use part of the metallacycles as the structure model for calculations. The authors should add comments on the rationality.

At present, it is difficult to accept this manuscript for publication without clear reply and answer to above comments and questions.

Reviewer #3 (Remarks to the Author):

Yang et al developed highly fluorescent coordination macrocycles and the use as fluorescent links. It is well known that coordination-driven self-assembly is a powerful tool to construct various 3D nanoarchitectures with functions. However, the reports on coordination rings, cages, and capsules with highly fluorescent ability are still limited. The authors newly designed and synthesized nine Pt-linked macrocycles with functionalized triphenylamine moieties. The triphenylamine-based PET and ICT properties change the fluorescent color and quantum yield of the coordination macrocycles. The structural and photophysical characters were well investigated by NMR, MS, UV-vis, fluorescence, lifetime, and theoretical analyses. The mechanistic discussion is reasonable. These results would give new knowledges to supramolecular chemistry and photochemistry. Therefore, I would like to strongly recommend publishing this paper in Nat. Commun. after the following minor revisions.

(i) There are other examples for fluorescent SCCs: J. Am. Chem. Soc. 2007, 129, 5300; J. Am. Chem. Soc. 2009, 131, 12526; Chem. Eur. J. 2012, 18, 8358; Chem. Sci. 2014, 5, 908; Chem. Eur. J. DOI: 10.1002/chem.201806409.

(ii) The quantum yields of macrocycles H1 and H4-6 are very high (more than 50%). The concentration of the compounds is quite important to estimate the fluorescent quantum yields. The information should be described in the main text and/or the figure/table captions.

(iii) The concentration-dependent fluorescent spectra and quantum yields of the representative macrocycles (e.g., 0.1, 1, 10, and 100 μM) can support the range of the utility. In addition, the possibility for the AIEE effect will be revealed.

(iv) The application for fluorescent links is very attractive. Is it possible to estimate the quantum yields of the obtained TLC plates and patterns of Fig. 8 and then compare them with the solution data? The detailed solid-state emissivity of the compounds is also important for further development.

Reviewer #1 (Remarks to the Author):

The authors presented a comprehensive study on the tuning of PET/ICT properties through altering the electron donating/accepting ability of the substituent of SCC systems. The fluorescence mechanism of the system has been explored and complexes with high fluorescence quantum yield have been achieved. However, there are problems.

1. Some mistakes have been spotted in the article. For example, in Scheme 2 (page 5), **L9** and **C9** were mistakenly denoted as **L10** and **C10** respectively. In Table 1 (page 5) and Table 2 (page 7), $-\text{CH}_3\text{O}$ should be $-\text{OCH}_3$.

Reply: We greatly appreciate the reviewer for pointing out the errors in the original manuscript. All errors have been corrected in the revised manuscript.

2.1 The article suggested the luminescence properties of **H1-H9** are fluorescence in nature (Page 5). They suggested that in some SCCs, the introduction of platinum center produces fluorescence enhancement due to PET inhibition, but in other SCCs fluorescence quenching due to heavy atom effect that gives significant ISC. The arguments are highly speculative and are not convincing. The authors did not discuss why in some SCCs the fluorescence prefer to be quenched and in others prefer to be enhanced. What are the driving forces for PET in these SCCs before Pt bonding? This can give information on the possibility of PET. There are no supporting evidences for the claims.

Reply: We fully understand the reviewer's concern about the role of coordination of platinum ions in determining the luminescence properties of metallacycles in this study. In order to make a clearer understanding, the basic concept of photoinduced electron transfer (PET) and intramolecular charge transfer (ICT) has been introduced in the Supporting Information (Supplementary Pages 3-5) as follows.

Photoinduced electron transfer (PET) is an excited state electron transfer process, in which an excited electron is transferred from donor to acceptor. As shown in Fig. R1, a typical PET molecule often includes three parts: a fluorophore that acts as the electron acceptor; a receptor that serves as an electron donor or a quencher; and a spacer that links the two parts of fluorophore and receptor. Fluorescent molecules based on PET are often structured as fluorophore-spacer-receptor constructs. In the PET system, the photoinduced electron transfer from the receptor to the fluorophore will induce fluorescence quenching. However, this photoinduced electron transfer (PET) process is restricted when the receptor binds upon its targets, which thus

induces the enhancement of fluorescence emission.

Fig. R1 Schematic representation of PET-based fluorescent system.

The mechanism of PET-based fluorescent molecule can be explained by the frontier orbital theory. As shown in Fig. R2a, the electron of the highest occupied molecular orbital (HOMO) is transferred to the lowest unoccupied molecular orbital (LUMO) when the fluorophore is excited by an appropriate light wavelength. Because the HOMO energy level of the receptor is between the LUMO and HOMO levels of the fluorophore, the electron transfers from the HOMO of the receptor to the HOMO of the fluorophore. This electron transfer induces the prevention of the electron in the LUMO of the fluorophore from returning to the HOMO, and thus leads to a fluorescence quenching by means of the PET effect. However, as shown in Fig. R1b and Fig. R2b, when the receptor binds upon the target, the HOMO energy of the receptor is declined to be lower than the HOMO energy level of the fluorophore. Therefore, the inhibition of PET process induces the enhancement of fluorescence emission.

Fig. R2 Frontier orbital theory for the PET effect.

Intramolecular charge transfer (ICT) is the charge transfer from an electron-rich donor moiety to an electron-poor acceptor part located in the same molecule. The ICT process generally occurs in the photoexcited state that a molecule reaches due to the absorption of light with an appropriate wavelength. The photoexcitation facilitates the transfer of an electron from one part of a molecule to its other part in the excited state, which leads to the charge distribution in the excited state markedly different from that in the ground state. Fluorescent molecules on the basis of ICT are featured by conjugation of an electron-donating unit to an electron-accepting unit in one

molecule to rise a “push–pull” π -electron system in the excited state (Fig. R3). When the electron-accepting part interacts with a guest, the electron-accepting character of the fluorescent molecule increases, thus generating a red shift in the emission spectrum (Fig. 3a and Fig. 3b). In contrast, an evident blue shift can be observed when the ICT becomes less developed due to the interaction of the electron-donating part with a guest (Fig. 3c and Fig. 3d).

Fig. R3 Schematic representation of ICT-based fluorescent systems.

The PET and ICT properties and mechanism of ligands **L1-L9** have been investigated by time-dependent density functional theory (TD-DFT) calculations, time-correlated single photon counting (TCSPC) method, and control experiments. All results provided solid evidences for our discussion in the manuscript. Moreover, in order to make it more reasonable, the detailed discussion has been added in the revised manuscript (Page 7) as follows.

For ligands **L2-L4**, which contained electron-withdrawing groups para to the tertiary amine core, their emission was mainly generated from the part of aniline and electron-withdrawing groups as witness by most of the electron densities of HOMO and LUMO were localized on the part of aniline and electron-withdrawing groups. Notably, ligand **L1** and metallacycle **H1** were exceptions, in which most of the electron densities were localized on the part of aniline and pyridine units although they were also substituted by electron-withdrawing group (trifluoromethyl), which might be caused by the well-known “fluorine effect”. In these ligands **L2-L4**, the part of aniline and electron-withdrawing groups acted as a fluorophore, pyridine units served as a receptor, and benzene units were the spacer that linked the two parts of fluorophore and receptor (Fig. R4). It indicated that they were PET molecules. Without binding of platinum(II), PET process occurred due to the transfer of electron density, originating at the lone pair electrons on N atoms of the pyridine units, to the

LUMO of the fluorophore. However, binding of platinum(II) to pyridine during the formation of metallacycles **H3-H4** hindered the PET process, thus leading to the enhancement of fluorescence intensity and quantum yield. Although the molecular simulation results also demonstrated that the binding of platinum(II) to pyridine during the formation of metallacycle **H2** hindered the PET process, both nitro group-containing metallacycle **H2** and ligand **L2** displayed extremely low quantum yields due to the strong $S1 \rightarrow T1$ intersystem crossing caused by nitro group. It should be noted that pyridine units in metallacycles **H2-H4** just served as a receptor rather than a fluorophore in this case. Therefore the heavy-atom effect arising from the platinum ion could be avoided even under the formation of platinum(II)-pyridyl SCCs.

For ligands **L5-L9**, which contained electron-donating groups para to the tertiary amine core, their emission was mainly generated from the part of aniline and pyridine units since the electron densities of HOMO of ligands **L5-L9** were distributed over the whole molecule while most of the electron densities of LUMO were localized on the part of aniline and pyridine units. In these ligands **L5-L9**, the part of aniline and pyridine units acted as a fluorophore, in which aniline and pyridine units served as donor and acceptor, respectively (Fig. R4). It means that ligands **L5-L9** were ICT molecules. Electron-donating groups such as $-H$, $-CH_3$, $-OCH_3$, $-NH_2$, and $-N(CH_3)_2$ could regulate the electron-donating ability of aniline, and then changed the emission wavelength of these fluorescent ligands. When the electron-accepting part pyridine units interacted with platinum(II), the electron-accepting character of the fluorescent molecule increases, thus generating a red shift in the emission spectrum. In the ligands **L5-L9**, pyridine units were important part of fluorophore and made large contribution to the fluorescence emission. The coordination of platinum(II) with pyridine during the formation of metallacycles **H5-H9** could induce the decrease of fluorescence intensity and quantum yield due to the heavy-atom effect arising from the platinum ion.

Fig. R4 Schematic illustrations of PET/ICT of ligands **L1-L9**.

2.2 Can the emission properties of H5-H9 be phosphorescence? Additional experiments should be done to support the assignments.

Reply: In order to confirm whether the emission properties of metallacycles **H5-H9** is fluorescence or phosphorescence, the delayed emission spectra of metallacycles **H5-H9** were measured. As shown in Fig. R5, very weak emission from metallacycles **H5-H7** and almost no emission from metallacycles **H8-H9** were observed. Moreover, the normalized steady-state emission spectra of metallacycles **H5-H7** were coincident with those of their delayed emission spectra (Fig. R6). Based on the Jablonski diagram (Fig. R7), phosphorescence emission from first triplet state (T1) is generally shifted to the longer wavelengths relative to the fluorescence due to its lower energy. Furthermore, transition from T1 to the singlet ground state is forbidden, and as a result the rate constants for triplet emission are several orders of magnitude smaller than those for fluorescence. However, the emission lifetime measurements of metallacycles **H5-H9** revealed that all of them were within nanosecond scale. The results further indicated that the emission properties of metallacycles **H5-H9** is fluorescence rather than phosphorescence.

Fig. R5 The emission spectra of metallacycles (a) **H5**, (b) **H6**, (c) **H7**, (d) **H8**, and (e) **H9** (2 μM in DCM, excitation at the absorbance maxima). Blue line: Steady-state emission spectra (delay time, 0 s). Red line: delayed emission spectra. (f) Delayed emission spectra of metallacycles **H5-H9** (2 μM in DCM, excitation at the absorbance maxima) in air. Delayed emission spectra were monitored in phosphorescence mode (total decay time, 20 ms; delay time, 0.1 ms; gate time, 2.0 ms).

Fig. R6 Normalized emission spectra of metallacycles (a) **H5**, (b) **H6**, (c) **H7** (2 μM in DCM, excitation at the absorbance maxima). Red line: steady-state emission spectra (delay time, 0 s). Blue line: delayed emission spectra. Delayed emission spectra were monitored in phosphorescence mode (total decay time, 20 ms; delay time, 0.1 ms; gate time, 2.0 ms).

Fig. R7 One form of a Jablonski diagram.

We have also added the results of delayed emission spectra in the supporting information (Supplementary Pages 99-100), and a brief discussion was added in the main text (Page 6) as follows:

*With the aim of confirming whether the emission properties of metallacycles **H5-H9** is fluorescence or phosphorescence, the delayed emission spectra of representative metallacycles **H5-H9** were measured. As shown in Fig. R5, very weak emission from metallacycles **H5-H7** and almost no emission from metallacycles **H8-H9** were observed. Moreover, the normalized steady-state emission spectra of metallacycles **H5-H7** were coincident with those of their delayed emission spectra (Fig. R6). Furthermore, the emission lifetime measurements of metallacycles **H5-H9** revealed*

*that all of them were within nanosecond scale. The results further indicated that the emission properties of metallacycles **H5-H9** is fluorescence rather than phosphorescence.*

3. Since phosphorescence can be quenched by oxygen, have the authors tried conducting the photophysical measurements under deoxygenated or oxygenated condition?

Reply: In the reply of question 2, it has been indicated that the emission properties of metallacycles **H5-H9** was fluorescence rather than phosphorescence. Just as the reviewer mentioned, the delayed fluorescence emission is sensitive to oxygen. Therefore, based on the reviewer's advice, the photophysical measurements including absorption spectra and the delayed fluorescence emission spectra of metallacycles **H5-H9** under deoxygenated or oxygenated condition were conducted. The spectral results and related discussion have been added in the supporting information (Supplementary Pages 100-101).

*As shown in Fig. R8, the absorption spectrum of metallacycles **H5-H9** tested under N_2 atmosphere was nearly the same as that tested in air, respectively. Moreover, the luminescence intensity of metallacycles **H5-H7** measured under N_2 atmosphere condition was slightly higher than that measured in air, respectively. In particular, the emission spectra of metallacycles **H8** and **H9** were found no significant changes between N_2 atmosphere and aerated atmosphere. The slight oxygen-sensitive delayed emission of metallacycles **H5-H7** could be considered as thermally activated delayed fluorescence from the first excited singlet state, which was obtained through the reverse intersystem crossing (RISC) process from oxygen-sensitive triplet excited state ($T1$) state.*

Fig. R8 Absorption spectra (a, c, e, g, i) and delayed emission spectra (b, d, f, h, k) of metallacycles **H5-H9** (2 μ M in DCM, excitation at the absorbance maxima) in air (blue line), and under N₂ atmosphere (red line). All detections were carried out in phosphorescence mode (total decay time, 20 ms; delay time, 0.1 ms; gate time, 2.0 ms).

4. The quantum yield (Page 5) should be specified as fluorescence quantum yield. They cannot be accurate up to two decimal places in % fluorescence QY. Are the absolute fluorescence QYs measured in solution or solid state? This is not specified. The SCCs are dynamic equilibrium mixtures. How to ensure the fluorescence QYs refer to the SCC or the mixture?

Reply: According to the reviewer's suggestion, the quantum yield has been specified as fluorescence quantum yield in the revised manuscript. Fluorescence quantum yields has been accurate to two significant digits in % fluorescence QY. In addition, the absolute fluorescence quantum yields were measured in solution. The related description has been added in the revised manuscript (Page 10) as follows.

Fluorescence quantum yields were measured in absolutely in solution using a commercial fluorometer with integrating sphere (RF-6000, shimadzu).

The fluorescence quantum yields of metallacycles **H1-H9** were measured in dichloromethane. The fluorescence quantum yields refer to the SCC rather than the mixture were ensured based on the following two reasons. Firstly, multi-wavelength fluorescence spectra of metallacycles **H1-H9** were investigated by three-dimensional excitation–emission matrix (3-D EEMs) technology. The results showed that the multi-wavelength fluorescence spectra of metallacycles **H1-H9** were consistent with their 1-D fluorescence spectra and no other light-emitting species was observed, which demonstrated that the metallacycles **H1-H9** displayed the independence of emission from excitation. Secondly, metallacycles **H1-H9** were constructed based on Pt–N coordination bond. Many previous reports have indicated that Pt–N coordination-based metallacycles displayed high stability in dichloromethane (*J. Am. Chem. Soc.*, **2017**, *139*, 9459).

5. Emission quantum yield under degassed media should be conducted to compare with the existing data. This is same for emission decay profiles. The interpretations of multiexponential processes are highly speculative with no supporting evidence.

Reply: Based on the Reviewer's suggestion, emission quantum yields and the emission decay profiles of ligands **L1-L9** and metallacycles **H1-H9** under the degassed media has been conducted (Fig. R9-12). As shown in Table R1, the fluorescence quantum yields of ligands **L1-L9** and metallacycles **H1-H9** tested in air were slightly lower than each of them measured under N₂ atmosphere, which might be

attributed to the collisional quenching effect of oxygen (*Appl. Phys. B*, **2005**, 80, 777). Moreover, most of fluorescence lifetimes of ligands and metallacycles in air were slightly shorter than those under N₂ atmosphere (Table R2-3), which might be also attributed to the quenching effect of oxygen resulting in an additional rate process that depopulates the excited state.

Table R1. Fluorescence quantum yields of ligands L1-L9 and metallacycles H1-H9 in DCM (5 μM) under N₂ atmosphere

Substituent	Ligand	Fluorescence quantum yield (%)	Metallacycle	Fluorescence quantum yield (%)
-CF ₃	L1	64	H1	71
-NO ₂	L2	1.0	H2	<1
-CH=C(CN) ₂	L3	15	H3	23
-CHO	L4	48	H4	62
-H	L5	61	H5	60
-CH ₃	L6	64	H6	56
-OCH ₃	L7	61	H7	10
-NH ₂	L8	2	H8	<1
-N(CH ₃) ₂	L9	<1.0	H9	<1

Fig. R9 Time-resolved fluorescence decay curves of ligands **L1** (a, at 420 nm), **L2** (b, at 450 nm), **L3** (c, at 560 nm), **L4** (d, at 485 nm), **L5** (e, at 440 nm), **L6** (f, at 450 nm), **L7** (g, at 475 nm), **L8** (h, at 410 nm), and **L9** (i, at 445 nm) in DCM (5 μ M) in air. Fitted by convoluting the IRF from the scattering of SiO₂ nanoparticles.

Fig. R10 Time-resolved fluorescence decay curves of metallacycles **H1** (a, at 520 nm), **H2** (b, at 580 nm), **H3** (c, at 590 nm), **H4** (d, at 520 nm), **H5** (e, at 520 nm), **H6** (f, at 540 nm), **H7** (g, at 520 nm), **H8** (h, at 500 nm), and **H9** (i, at 520 nm) in DCM (5 μ M) in air. Fitted by convoluting the IRF from the scattering of SiO₂ nanoparticles.

Fig. R11 Time-resolved fluorescence decay curves of ligands **L1** (a, at 420 nm), **L2** (b, at 450 nm), **L3** (c, at 560 nm), **L4** (d, at 485 nm), **L5** (e, at 440 nm), **L6** (f, at 450 nm), **L7** (g, at 475 nm), **L8** (h, at 410 nm), and **L9** (i, at 445 nm) in DCM (5 μ M) under N_2 atmosphere. Fitted by convoluting the IRF from the scattering of SiO_2 nanoparticles.

Fig. R12 Time-resolved fluorescence decay curves of metallacycles **H1** (a, at 520 nm), **H2** (b, at 580 nm), **H3** (c, at 590 nm), **H4** (d, at 520 nm), **H5** (e, at 520 nm), **H6** (f, at 540 nm), **H7** (g, at 520 nm), **H8** (h, at 500 nm), and **H9** (i, at 520 nm) in DCM (5 μ M) under N_2 atmosphere. Fitted by convoluting the IRF from the scattering of SiO_2 nanoparticles.

The fluorescence quantum yields of ligands **L1-L9** and metallacycles **H1-H9** under N_2 atmosphere, time-resolved fluorescence decay curves and fluorescence lifetimes of ligands **L1-L9** and metallacycles **H1-H9**, and the related brief discussion have been added in the supporting information (Supplementary Page 83, 91-94) as follows.

*As shown in Table R1, the fluorescence quantum yields of ligands **L1-L9** and metallacycles **H1-H9** tested in air were slightly lower than each of them measured under N_2 atmosphere, which might be attributed to the collisional quenching effect of*

oxygen. Moreover, most of fluorescence lifetimes of ligands and metallacycles in air were slightly shorter than those under N₂ atmosphere (Table R2-3), which might be also attributed to the quenching effect of oxygen resulting in an additional rate process that depopulates the excited state.

The fluorescence lifetimes of ligands **L1-L9** and metallacycles **H1-H9** were measured based on time-correlated single-photon counting (TCSPC) technique. The squared deviations obtained by the least-squares analysis was utilized to estimate the fitting of TCSPC data and judge the multiexponential emission decay processes.

In order to make a clearer understanding, the related description on the measurement of the fluorescence lifetimes of ligands **L1-L9** and metallacycles **H1-H9** has been added in the Method section in the main text (Page 10) as follows.

Fluorescence lifetimes. *The fluorescence lifetimes were measured based on time-correlated single-photon counting (TCSPC) technique. The squared deviations obtained by the least-squares analysis was utilized to estimate the fitting of TCSPC data and judge the multiexponential emission decay processes.*

Table R2 The fluorescence lifetimes of ligands L1-L9 in DCM under N₂ atmosphere

Ligand		1	2	3	chisqr
L1	Lifetime/ns	0.3	1.0	1.8	1.07
	Percent/%	17	7	76	
	Average/ns	1.7			
L2		1	2		chisqr
	Lifetime/ns	1.5	4.8		1.07
	Percent/%	83	17		
	Average/ns	2.8			
L3		1	2		chisqr
	Lifetime/ns	1.8	6.2		1.08
	Percent/%	95	5		
	Average/ns	2.4			
L4		1	2		chisqr
	Lifetime/ns	2.1	6.4		1.02
	Percent/%	31	69		
	Average/ns	5.9			
L5		1	2		chisqr
	Lifetime/ns	0.2	2.4		1.06
	Percent/%	40	60		
	Average/ns	2.2			
L6		1	2		chisqr
	Lifetime/ns	0.3	2.9		1.02
	Percent/%	30	70		
	Average/ns	2.8			
L7		1	2	3	chisqr
	Lifetime/ns	0.1	1.4	4.5	1.04
	Percent/%	48	3	49	
	Average/ns	4.3			
L8		1	2		chisqr
	Lifetime/ns	2.0	5.9		1.11
	Percent/%	96	4		
	Average/ns	2.4			
L9		1	2	3	chisqr
	Lifetime/ns	0.1	1.9	3.3	1.06
	Percent/%	55	22	23	
	Average/ns	2.7			

Table R3 The fluorescence lifetimes of metallacycles H1-H9 in DCM under N₂ atmosphere

Metallacycle		1	2		chisqr
H1	Lifetime/ns	1.4	2.6		1.07
	Percent/%	22	78		
	Average/ns	2.4			
H2		1	2	3	chisqr
	Lifetime/ns	0.1	0.8	1.9	0.96
	Percent/%	92	3	5	
	Average/ns	0.9			
H3		1	2	3	chisqr
	Lifetime/ns	0.4	2.0	2.8	1.07
	Percent/%	15	61	24	
	Average/ns	2.2			
H4		1	2		chisqr
	Lifetime/ns	1.7	2.8		1.06
	Percent/%	45	55		
	Average/ns	2.4			
H5		1	2	3	chisqr
	Lifetime/ns	0.4	2.3	3.6	1.08
	Percent/%	24	32	44	
	Average/ns	3.1			
H6		1	2	3	chisqr
	Lifetime/ns	0.5	2.5	3.8	1.05
	Percent/%	23	26	51	
	Average/ns	3.3			
H7		1	2	3	chisqr
	Lifetime/ns	0.2	0.7	3.3	1.12
	Percent/%	32	67	1	
	Average/ns	0.9			
H8		1	2	3	chisqr
	Lifetime/ns	0.2	1.4	3.3	1.04
	Percent/%	34	50	16	
	Average/ns	2.1			
H9		1	2	3	chisqr
	Lifetime/ns	0.3	1.3	3.3	1.07
	Percent/%	46	36	18	
	Average/ns	2.2			

6. The article demonstrated the utilization of the complexes as fluorescent ink for printing (page 8-9). It is suggested that comments should be added to emphasize the advantages of using metallacycles over simple metal complexes (eg. complex C3, C4) or organic fluorescence ligands (eg. ligand L5, L6) for such purpose.

Reply: According to the reviewer's suggestion, the utilization of model complexes **C3-C4** and the organic fluorescence ligands **L5-L6** as fluorescent inks for printing fluorescent patterns has been conducted. As shown in Fig. R13, all of the patterns printed from complexes **C3**, **C4** as well as ligands **L5**, **L6** solutions were invisible under the ambient conditions. Moreover, these printed patterns of the Chinese knot with yellow fluorescence, blue-green fluorescence, green fluorescence, and cyan fluorescence, respectively, could be observed clearly under the excitation of hand-held UV lamp (365 nm).

It should be noted that the patterns printed from metallacycles displayed different fluorescence colors compared to the patterns from simple model metal complexes and organic fluorescence ligands. This finding further illustrated that PET/ICT regulation strategy was an effective approach to construct organoplatinum metallacycles with tunable fluorescence wavelengths not only in solution but also in solid state.

Fig. R13 Photographs of the fluorescent patterns by inkjet printing from complex **C3** in visible light (a) and under 365 nm light excitation (e), complex **C4** in visible light (b) and under 365 nm light excitation (f), ligand **L5** in visible light (c) and under 365 nm light excitation (g), and ligand **L6** in visible light (d) and under 365 nm light excitation (h).

We have also added the photographs of the fluorescent patterns by inkjet printing from complexes **C3-C4** and ligands **L5-L6** into the supporting information (Supplementary Page 103), and a brief discussion has been added in the main text (Pages 9-10) as follows:

Moreover, it should be noted that the patterns printed from metallacycles displayed different fluorescence colors compared to the patterns from simple model metal complexes and organic fluorescence ligands. This finding further illustrated that PET/ICT regulation strategy was an effective approach to construct organoplatinum metallacycles with tunable fluorescence wavelengths not only in solution but also in solid state.

7. The authors suggested that many organic materials exhibit intense emission only in solution state (page 8). To make the study more comprehensive, it is suggested that some precursor ligands should be selected to investigate their solid-state emission properties (fluorescent films / ink printing).

Reply: Based on the reviewer's suggestion, precursor ligands **L1**, **L3**, and **L7** were selected as the representatives to investigate their solid-state emission properties including fluorescent films and ink printing. As shown in Fig. R14, 3-D fish-shape fluorescent films and the printed patterns of the Chinese knot fabricated from ligands **L1**, **L3**, and **L7** exhibited weak fluorescence with different colors. For example, 3-D fish-shape fluorescent films and the printed patterns of the Chinese knot prepared from ligands **L1** displayed green fluorescence. It should be noted that the fluorescent films and printed patterns made from metallacycle **H1** emitted blue fluorescence with the stronger emission. These results also demonstrated that PET/ICT regulation strategy was an effective method for preparing organoplatinum metallacycles with tunable quantum yields and fluorescence wavelengths.

Fig. R14 Photographs of the flexible fluorescent films (a-f) and the fluorescent patterns (g-l) by inkjet printing from ligand **L1** in visible light (a, g) and under 365 nm light excitation (d, j), ligand **L3** in visible light (b, h) and under 365 nm light excitation (e, k), ligand **L7** in visible light (c, i) and under 365 nm light excitation (f, l).

We have also added the photographs of the flexible fluorescent films and the fluorescent patterns by inkjet printing from ligands **L1**, **L3**, and **L7** into the supporting

information (Supplementary Page 103), and a brief discussion has been added in the main text (Page 9) as follows:

*Moreover, fluorescent films fabricated from metallacycles and ligands exhibited different fluorescence properties. For example, 3-D fish-shape fluorescent films prepared from ligands **LI** displayed green fluorescence while the fluorescent films made from metallacycle **HI** emitted blue fluorescence with the stronger emission.*

The novelty of the work is thin and is not up to standard for publication in Nature Comm.

Reply: We don't agree with this reviewer's opinion on the novelty of this work.

During the past few years, the preparation of fluorescent discrete SCCs has garnered great attention due to their promising applications such as chemical sensors, optical devices, supramolecular biomedicines, and so on. Moreover, the presence of chromophores in SCCs allows for real-time monitoring the self-assembly process and dynamics of the resultant SCCs by highly sensitive fluorescence technique. Although the fruitful achievements have been gained in the development of fluorescent discrete SCCs, many problems are still far from being fully resolved in this area. For example, the heavy-atom effect arising from the Pt(II)-based building block sometimes induces a significant increase in the amounts of intersystem crossing (ISC) due to spin-orbit coupling, which thus leads to an obvious fluorescence quenching of the discrete SCCs. Secondly, there is a relative lack of efficient and simple strategy to tune the emission wavelength of the discrete SCCs. Up to now, changing the solvents is the mostly used method to tune the emission wavelength of the discrete SCCs. However, some solvents such as acetonitrile could disrupt the Pt–N coordination bond because these solvents feature the stronger binding ability to platinum atom than pyridine. Therefore, it is still challenging to construct discrete organoplatinum SCCs with high quantum yields and tunable fluorescence emission wavelengths.

In this work, we presented the first example on the construction of organoplatinum metallacycles with high quantum yields and tunable fluorescence wavelengths by simply switching the PET and ICT properties of building blocks based on the substituent effect. This PET/ICT switchable strategy not only offers a good guidance on how to avoid the heavy-atom effect arising from the platinum ion in the construction of fluorescent platinum(II)-pyridyl SCCs, but also provides an alternative approach to prepare fluorescent platinum(II)-pyridyl SCCs with high quantum yields and even tunable wavelength of the discrete SCCs without changing the solvents. The photophysical properties of prepared metallacycles **H1-H9**, including absorption and

2-D fluorescence emission spectra, 3-D excitation-emission matrix spectra, fluorescence quantum yields, and fluorescence lifetimes, have been comprehensively investigated. The luminescence mechanisms of these metallacycles **H1-H9** have been systematically investigated by TCSPC method, TD-DFT calculation, and the control experiments. Moreover, by utilizing the excellent fluorescence properties of resultant metallacycles, many kinds of fluorescent materials including fluorescent metallacycle-loaded thin-layer chromatography (TLC) plates, 3-D-fish-shape fluorescent films, and fluorescent inks for inkjet printing were successfully prepared, which displayed wide potential applications in anti-counterfeiting, information hiding and storage, and optoelectronic devices.

More importantly, as the other two reviewers commented, the reported results are interesting and these results would give new knowledges to supramolecular chemistry and photochemistry. Thus, we believe that the chemistry reported in the manuscript should has enough novelty and be interesting to a broad readership of *Nat. Commun.*

In order to gain a better understanding and highlight the novelty of the chemistry presented in this manuscript, major improvement especially the detailed discussion on the basic concept of PET and ICT has been made in the introduction section as follows.

A typical PET molecule often includes three parts: a fluorophore that acts as the electron acceptor; a receptor that serves as an electron donor or a quencher, and a spacer that links the two parts of fluorophore and receptor. Fluorescent molecules based on PET are often structured as fluorophore-spacer-receptor constructs. In the PET system, the photoinduced electron transfer from the receptor to the fluorophore will induce fluorescence quenching. However, this photoinduced electron transfer (PET) process is restricted when the receptor binds upon its targets, which thus induces the enhancement of fluorescence emission. However, fluorescent molecules on the basis of ICT are featured by conjugation of an electron-donating unit to an electron-accepting unit in one molecule to rise a “push–pull” π -electron system in the excited state. When the electron-accepting part interacts with a guest, the electron-accepting character of the fluorescent molecule increases, thus generating a red shift in the emission spectrum. In contrast, an evident blue shift can be observed when the ICT becomes less developed due to the interaction of the electron-donating part with a guest.

In this work, we presented the first example on the preparation of organoplatinum metallacycles with high quantum yields and tunable fluorescence wavelengths by

simply switching the PET and ICT properties of building blocks based on the substituent effect. This PET/ICT switchable strategy not only offers a good guidance on how to avoid the heavy-atom effect arising from the platinum ion in the construction of fluorescent platinum(II)-pyridyl SCCs, but also provides an alternative approach to prepare fluorescent platinum(II)-pyridyl SCCs with high quantum yields and even tunable wavelength of the discrete SCCs without changing the solvents.

Reviewer #2 (Remarks to the Author):

In this manuscript, the authors report the systematic construction of organoplatinum metallacycles with high quantum yields and tunable fluorescence wavelengths by switching the PET and ICT properties of the building blocks. Detailed controlled experiments and DFT/TDDFT calculations were performed to give a further insight into the photophysical mechanism. Although the reported results are interesting, this manuscript has several issues as listed below:

1. For the comments “the observed results were attributed to the coordination of metal center with the pyridyl nitrogen, which enriched the ligand π -system, lowered the energy required for excitation, and helped to stabilize the LUMO” in the manuscript, a relevant reference should be provided.

Reply: According to the reviewer’s suggestion, the relevant reference (*J. Am. Chem. Soc.*, **2012**, *134*, 10607) has been provided in the revised manuscript.

2. I feel confused for the fluorescence quantum yield of H2. As shown in Table 1, the quantum yield of L2 is 2.14%, while the quantum yield of H2 is 0.6%. However, the author claim that “binding of platinum(II) to pyridine during the formation of metallacycles H2-H4 hindered the PET process, thus leading to the enhancement of fluorescence intensity and quantum yield”. Is there a mistake?

Reply: We fully understand the reviewer’s concern about the quantum yield of metallacycle **H2**. The theoretical molecular simulation of metallacycle **H2** by using time-dependent density functional theory (TD-DFT) was performed, which indicated that the binding of platinum (II) to pyridine during the formation of metallacycle **H2** hindered the PET process, thus leading to the enhancement of fluorescence intensity and quantum yield. However, as indicated by the experimental results, both nitro group-containing metallacycle **H2** and ligand **L2** displayed extremely low quantum yields (~1% for **H2** and ~2% for **L2**), which might be attributed to the strong S1→T1

intersystem crossing caused by nitro group (*J. Phys. Chem. A*, **2008**, *112*, 6313; *J. Phys. Chem. A*, **2013**, *117*, 6580). In order to avoid the possible misunderstanding and make it more reasonable, the related discussion has been modified as follows.

Binding of platinum(II) to pyridine during the formation of metallacycles H3-H4 hindered the PET process, thus leading to the enhancement of fluorescence intensity and quantum yield. Although the molecular simulation results also demonstrated that the binding of platinum(II) to pyridine during the formation of metallacycle H2 hindered the PET process, both nitro group-containing metallacycle H2 and ligand L2 displayed extremely low quantum yields due to the strong S1→T1 intersystem crossing caused by nitro group.

3. The authors point out that the coordination of metallacycles H5-H9 can induce the quantum yield due to the heavy atom effect arising from the platinum ion. Accordingly, nonradiative rate constants k_{nr} will be increased. But for the data of H5-H8 as shown in table 2, why is their k_{nr} not larger than that of the corresponding ligands?

Reply: We fully understand the reviewer's concern about the relationship between fluorescence quantum yield and nonradiative rate constant k_{nr} . As the reviewer mentioned, the nonradiative rate constants (k_{nr}) of metallacycles **H5-H8** were not larger than those of their corresponding ligands. However, the value of the ratio nonradiative rate constant (k_{nr})/radiative rate constant (k_{rad}) of each metallacycle (**H5-H8**) was larger than that of the corresponding ligand, which provided a quantitative understanding of the lower quantum yields of metallacycles **H5-H8** compared to their constituent ligands **L5-L8**. With the aim of better understanding, the related discussed has been added in the revised manuscript (Page 6) as follows.

Although the nonradiative rate constants (k_{nr}) of metallacycles H5-H8 were not larger than those of their corresponding ligands, the value of the ratio nonradiative rate constant (k_{nr})/radiative rate constant (k_{rad}) of each metallacycle (H5-H8) was larger than that of the corresponding ligand, which provided a quantitative understanding of the lower quantum yields of metallacycles H5-H8 compared to their constituent ligands L5-L8.

4. The authors use part of the metallacycles as the structure model for calculations. The authors should add comments on the rationality.

Reply: We agree with the reviewer's opinion on molecular simulation. In order to provide the more convincing molecular calculations, ligands **L3** and **L7** as well as metallacycles **H3** and **H7** were selected as the representatives to investigate the whole

molecular simulation through the analysis of hole and electron distribution by using time-dependent density functional theory (TD-DFT).

We have added the results into the supporting information (Supplementary Pages 115-117), and a brief discussion has been added in the main text (Pages 7-8) as follows:

*In order to provide the more convincing molecular calculations, ligands **L3** and **L7** as well as metallacycles **H3** and **H7** were selected as the representatives to investigate the whole molecular simulation through the analysis of hole and electron distribution by using time-dependent density functional theory (TD-DFT). As shown in Fig. R15, for ligand **L3**, holes and electrons were mainly located at the part of aniline and electron-withdrawing groups, which indicated that electrons transferred from the aniline groups to the electron-withdrawing groups when ligand **L3** was excited to the S1 state. However, for ligand **L7**, holes and electrons were distributed at the part of aniline and pyridine units (Fig. R16). This result showed that electrons transferred from the aniline groups to the pyridine units when ligand **L7** was excited to the S1 state. For metallacycle **H3**, the hole-electron distributions of S1, S2, S3 states, which derived from three ligands respectively, were similar with each other (Fig. R17). More importantly, all of them were similar with the hole-electron distributions of S1 states of the corresponding ligand **L3**. These results suggested that electrons also transferred from the aniline groups to the pyridine units when excitation of metallacycle **H3**. Similar phenomena were also observed in metallacycle **H7** (Fig. R18).*

Fig. R15 Electron-hole distribution of the S1 state for ligand **L3**. (a) The blue colors represent holes; (b) The green colors represent electrons; (c) Overlap of electron-hole for the S1 state of ligand **L3**.

Fig. R16 Electron-hole distribution of the S1 state for ligand **L7**. (a) The blue colors represent holes; (b) The green colors represent electrons; (c) Overlap of electron-hole for the S1 state of ligand **L7**.

Fig. R17 Electron-hole distribution of the (a-c) S1, (d-f) S2, and (g-i) S3 states for metallacycle **H3**. The blue colors represent holes, the green colors represent electrons. Overlap of electron-hole for the (c) S1, (f) S2, and (i) S3 states of metallacycle **H3**.

Fig. R18 Electron-hole distribution of the (a-c) S1, (d-f) S2, and (g-i) S3 states for metallacycle **H7**. The blue colors represent holes, the green colors represent electrons. Overlap of electron-hole for the (c) S1, (f) S2, and (i) S3 states of metallacycle **H7**.

At present, it is difficult to accept this manuscript for publication without clear reply and answer to above comments and questions.

Reply: According to the reviewers' thorough and constructive suggestion, many supplementary experiments and comprehensive modifications have been conducted, which obviously improved the quality of this manuscript. Therefore, we hope the revised manuscript is now acceptable for publication in *Nat. Commun.*

Reviewer #3 (Remarks to the Author):

Yang et al developed highly fluorescent coordination macrocycles and the use as fluorescent links. It is well known that coordination-driven self-assembly is a powerful tool to construct various 3D nanoarchitectures with functions. However, the reports on coordination rings, cages, and capsules with highly fluorescent ability are still limited. The authors newly designed and synthesized nine Pt-linked macrocycles with functionalized triphenylamine moieties. The triphenylamine-based PET and ICT properties change the fluorescent color and quantum yield of the coordination macrocycles. The structural and photophysical characters were well investigated by NMR, MS, UV-vis, fluorescence, lifetime, and theoretical analyses. The mechanistic discussion is reasonable. These results would give new knowledges to supramolecular chemistry and photochemistry. Therefore, I would like to strongly recommend publishing this paper in Nat. Commun. after the following minor revisions.

Reply: We greatly appreciate the reviewer for his/her positive comments on our manuscript.

(i) There are other examples for fluorescent SCCs: J. Am. Chem. Soc. 2007, 129, 5300; J. Am. Chem. Soc. 2009, 131, 12526; Chem. Eur. J. 2012, 18, 8358; Chem. Sci. 2014, 5, 908; Chem. Eur. J. DOI: 10.1002/chem.201806409.

Reply: All literatures mentioned above have been cited in the revised manuscript.

(ii) The quantum yields of macrocycles H1 and H4-6 are very high (more than 50%). The concentration of the compounds is quite important to estimate the fluorescent quantum yields. The information should be described in the main text and/or the figure/table captions.

Reply: We fully agree with the reviewer's opinion that the concentration of the compounds is quite important to estimate the fluorescent quantum yields. Therefore, the related information has been described in the revised main text.

(iii) The concentration-dependent fluorescent spectra and quantum yields of the representative macrocycles (e.g., 0.1, 1, 10, and 100 microM) can support the range of the utility. In addition, the possibility for the AIEE effect will be revealed.

Reply: According to the reviewer's suggestion, metallacycle **H6** was selected as the representative to investigate the concentration-dependent fluorescent spectra and quantum yields.

The results and a brief discussion have been added into the supporting information (Supplementary Page 84) as follows:

As shown in Fig. R19, upon increasing the concentration of metallacycle **H6** from 0.5 μM to 10 μM , the significant increases both in absorption spectra and fluorescent emission spectra were observed. However, as the concentration increased, metallacycle **H6** exhibited a small change in the fluorescence quantum yield (Table R4). These results indicated that metallacycle **H6** didn't display the obvious AIEE effect.

Fig. R19 Absorption and fluorescent emission spectra of metallacycle **H6** in CH_2Cl_2 at different concentrations.

Table R4 The fluorescence quantum yields of metallacycle **H6** in CH_2Cl_2 at different concentrations

Concentration	0.5 μM	1.0 μM	3.0 μM	5.0 μM	10.0 μM
Fluorescence quantum yield (%)	53	50	53	52	51

(iv) The application for fluorescent inks is very attractive. Is it possible to estimate the quantum yields of the obtained TLC plates and patterns of Fig. 8 and then compare them with the solution data? The detailed solid-state emissivity of the compounds is also important for further development.

Reply: We greatly appreciate the reviewer for his/her positive comments on the fluorescent inks of the prepared fluorescent materials. We fully agree with the reviewer's opinion that the detailed solid-state emission properties including fluorescence quantum yield are very important for further development. Therefore, the absorption spectra, fluorescent emission spectra, and fluorescence quantum yields of **H1**, **H3**, and **H7**-functionalized polymer films were investigated as shown in Fig. R20-22. **H1**-functionalized polymer film ($\lambda_{\text{em}} = 508 \text{ nm}$) displayed blue-shifted emission when being compared with metallacycle **H1** in solution ($\lambda_{\text{em}} = 487 \text{ nm}$). Moreover, both **H3** and **H7**-functionalized polymer films ($\lambda_{\text{em}} = 576 \text{ nm}$ for **H3**, $\lambda_{\text{em}} = 516 \text{ nm}$ for **H7**) displayed an obvious blue-shifted emission when being compared with metallacycles **H3** and **H7** in solution ($\lambda_{\text{em}} = 586 \text{ nm}$ for **H3**, $\lambda_{\text{em}} = 555 \text{ nm}$ for **H7**). It might be attributed to the different aggregation behaviors of metallacycles **H1**,

H3, and **H7** within the film. Notably, all of them exhibited moderate fluorescence quantum yields of 28%, 18%, and 23% for **H1**, **H3**, and **H7** respectively.

Fig. R20 Absorption (left), emission (right, $\lambda_{\text{exc}} = 393$ nm) spectra of thin film doped with 0.1 wt% of metallacycle **H1**.

Fig. R21 Absorption (left), emission (right, $\lambda_{\text{exc}} = 442$ nm) spectra of thin film doped with 0.1 wt% of metallacycle **H3**.

Fig. R22 Absorption (left), emission (right, $\lambda_{\text{exc}} = 413$ nm) spectra of thin film doped with 0.1 wt% of metallacycle **H7**.

Unfortunately, all attempts to investigate the detailed solid-state emission properties of obtained TLC plates and fluorescent patterns of the Chinese knot have proven to be unsuccessful due to the equipment limitation. However, it was found that the prepared metallacycles-loaded TLC plates emitted different color fluorescence upon excitation under a hand-held UV lamp and the fluorescent patterns displayed “vis-invisible” and

“UV-visible” properties with multicolor fluorescence, which may allow for their potential applications in anti-counterfeiting, information hiding and storage, and optoelectronic devices.

The related results and discussion have been shown in Supplementary Page 102 and in main text (Page 9).

REVIEWERS' COMMENTS:

Reviewer #1 (Remarks to the Author):

The authors have made an effort to perform additional experiments and to make amendments to the manuscript to address most of the questions concerned. There are still some points that need to be addressed.

1 The basic concepts of PET and ICT have been included in both supporting information and the introduction of the manuscript. The authors use the term "guest" to represent the general phenomenon of binding a target molecule in lowering the HOMO energy of the receptor unit, which is likely to be inappropriate as the electron nature of the "guest" is not specified in the illustration. Some comments should be included to describe the nature of the "guest" in order to achieve PET in the current illustration. For example, it is unlikely to have an electron-donating guest for such an observation.

2 Similarly, the nature of the "guest" for the general description of the ICT process also needs to be specified. The electron-affinity nature of the guest is likely to affect the redshifting or blueshifting of the emission properties of the current system.

3 The authors did not estimate the driving force for these processes. They can be readily estimated from thermodynamic data and should be included.

4 The authors have attributed the low emission quantum yield of H2 and L2 to an ISC process. Some more comments are needed to briefly explain this phenomenon.

Reviewer #2 (Remarks to the Author):

The authors have addressed most of the reviewers' comment. However, I comprehend that there are quite a number of the recent refs for supramolecular coordination complexes studies that could not be cited, e.g., Direct Observation of a Triplet-State Absorption-Emission Conversion in a Fullerene-Functionalized Pt(II) Metallacycle with femtosecond transient absorption experiments and quantum chemistry calculations (JOURNAL OF PHYSICAL CHEMISTRY C 121 (27): 14975-14980, 2017); Unveiling excited state energy transfer and charge transfer in a host/guest coordination cage by combination of femtosecond transient absorption spectroscopy, nanosecond transient emission spectroscopy and quantum chemistry calculations (PHYSICAL CHEMISTRY CHEMICAL PHYSICS 20 (4): 2205-2210 , 2018)

Reviewer #3 (Remarks to the Author):

Yang et al clearly answered all of my questions and comments on the basis of their additional experiment data. Now, the novelty, importance and quality of this work adequately reached to this journal. Therefore, I would like to strongly recommend publishing this paper in Nat. Commun. soon.

Reviewer #1 (Remarks to the Author):

The authors have made an effort to perform additional experiments and to make amendments to the manuscript to address most of the questions concerned. There are still some points that need to be addressed.

1. The basic concepts of PET and ICT have been included in both supporting information and the introduction of the manuscript. The authors use the term “guest” to represent the general phenomenon of binding a target molecule in lowering the HOMO energy of the receptor unit, which is likely to be inappropriate as the electron nature of the “guest” is not specified in the illustration. Some comments should be included to describe the nature of the “guest” in order to achieve PET in the current illustration. For example, it is unlikely to have an electron-donating guest for such an observation.

Reply: We fully understand the reviewer’s concern about the illustration of the basic concept of PET and ICT. In order to clear up the possible confusion, the term of “guest” has been specified in the illustration and the related description and discussion has been modified in the revised manuscript as follows.

However, this photoinduced electron transfer (PET) process is restricted when the receptor binds upon its electron-withdrawing targets (such as metal ions), which thus induces the enhancement of fluorescence emission.

However, as shown in Supplementary Figure 1b and Supplementary Figure 2b, when the receptor binds upon the electron-withdrawing target, the HOMO energy of the receptor is declined to be lower than the HOMO energy level of the fluorophore.

2. Similarly, the nature of the “guest” for the general description of the ICT process also needs to be specified. The electron-affinity nature of the guest is likely to affect the redshifting or blueshifting of the emission properties of the current system.

Reply: Similarly, in order to make a better understanding, the term of “guest” has been specified in the illustration and the related description and discussion of the ICT process has been modified in the revised manuscript as follows.

When the electron-accepting part interacts with an electron-withdrawing guest (such as metal ions), the electron-accepting character of the fluorescent molecule increases, thus generating a red shift in the emission spectrum. In contrast, an evident blue shift can be observed when the ICT becomes less developed due to the interaction of the electron-donating part with an electron-withdrawing guest.

3. The authors did not estimate the driving force for these processes. They can be

readily estimated from thermodynamic data and should be included.

Reply: We fully understand the reviewer's concern about the driving force for PET process in these metallacycles. According to the reviewer's suggestion, we have tried to measure the oxidation potential of electron donor and reduction potential of electron acceptor/fluorophore in metallacycles **H1**, **H3**, and **H4** to obtain thermodynamic data, which has proven unsuccessful. However, in order to provide more information about the PET process, the additional control experiment was conducted. Now, the PET process from pyridine to substituted triarylamine in metallacycles **H1**, **H3**, and **H4** has been accurately confirmed by absorption and fluorescence emission spectra, TCSPC method, TD-DFT calculation, as well as the control experiment. The related discussion has been added in the revised manuscript as follows.

*Moreover, in order to ensure that the enhancement of fluorescence for metallacycles **H1**, **H3**, and **H4** were caused by the inhibition of PET process through the platinum(II)-pyridyl coordination, a representative control experiment was conducted by mixing dipyridyl ligand **L3** and platinum–bromine complex **11** for 6.0 hours (Fig. R1). The platinum atoms in complex **11** were protected by bromine moieties, which cannot coordinate with ligand **L3** to form a metallacycle. As shown in Fig. R2, almost no change in the absorption and emission spectra of ligand **L3** was observed, which further demonstrated that both reasons for inhibiting PET process and the driving force for metallacycle formation were the platinum(II)-pyridyl coordination.*

Fig. R1 The stirring of ligand **L3** and compound **11** in dichloromethane for 6.0 hours cannot form the metallacycle.

Fig. R2 Time-dependent changes in the absorption (a) and emission (b) spectra of the mixture of ligand **L3** and compound **11** in dichloromethane.

4. The authors have attributed the low emission quantum yield of **H2** and **L2** to an ISC process. Some more comments are needed to briefly explain this phenomenon.

Reply: According to the reviewer's suggestion, more detailed comments have been added to explain the low emission quantum yields of **H2** and **L2** as follows.

*Although metallacycle **H2** contains electron-withdrawing group (nitro-group) para to the tertiary amine core, metallacycle **H2** displayed an extreme low fluorescence quantum yield (~1%) similar to that of its constituent ligand **L2** (~2%). This observation was attributed to the presence of nitro-groups in the triarylamine skeleton of **H2** and **L2** that caused the usual decrease of radiative deactivation in the singlet state in favour of significant intersystem crossing to the triplet state, thus quenching the fluorescence. The similar nitro-group effect has been previously reported in literatures⁵⁵*

Reviewer #2 (Remarks to the Author):

The authors have addressed most of the reviewers' comment. However, I comprehend that there are quite a number of the recent refs for supramolecular coordination complexes studies that could not be cited, e.g., Direct Observation of a Triplet-State Absorption-Emission Conversion in a Fullerene-Functionalized Pt(II) Metallacycle with femtosecond transient absorption experiments and quantum chemistry calculations (JOURNAL OF PHYSICAL CHEMISTRY C 121 (27): 14975-14980, 2017); Unveiling excited state energy transfer and charge transfer in a host/guest coordination cage by combination of femtosecond transient absorption spectroscopy, nanosecond transient emission spectroscopy and quantum chemistry calculations (PHYSICAL CHEMISTRY CHEMICAL PHYSICS 20 (4): 2205-2210 , 2018)

Reply: The suggested references have been cited in the revised manuscript.

Reviewer #3 (Remarks to the Author):

Yang et al clearly answered all of my questions and comments on the basis of their additional experiment data. Now, the novelty, importance and quality of this work adequately reached to this journal. Therefore, I would like to strongly recommend publishing this paper in Nat. Commun. soon.

Reply: We greatly appreciate the reviewer for his/her positive comments on our manuscript.